# Multipolar condensates and multipolar Josephson effects

**Wenhui Xu** [1], **Chenwei Lv**[1] **& Qi Zhou** [1,2] ✉

When single-particle dynamics are suppressed in certain strongly correlated systems, dipoles arise as elementary carriers of quantum kinetics. These dipoles can further condense, providing physicists with a rich realm to study fracton phases of matter. Whereas recent theoretical discoveries have shown that an unconventional lattice model may host a dipole condensate as the ground state, we show that dipole condensates prevail in bosonic systems due to a self-proximity effect. Our findings allow experimentalists to manipulate the phase of a dipole condensate and deliver dipolar Josephson effects, where supercurrents of dipoles arise in the absence of particle flows. The self-proximity effects can also be utilized to produce a generic multipolar condensate. The kinetics of the $n$-th order multipoles unavoidably creates a condensate of the $(n+1)$-th order multipoles, forming a hierarchy of multipolar condensates that will offer physicists a whole new class of macroscopic quantum phenomena.

The pursuit of new quantum matter has been the main theme in modern physics. When interactions produce profound correlations in a many-body system, novel quantum phases arise. Exemplary such phases include but are not limited to superfluids in Helium[1], high-Tc superconductors[2], and quantum Hall states[3]. Recent studies have found a new class of quantum matter, which is constituted by fractons[4–14]. Distinct from conventional particles, fractons are immobile as the movement of a single fracton may create extra excitations and thus an additional energy penalty. Bound states of fractons, however, can move and are responsible for the kinetics of the system. The fracton phase of matter has intrigued enthusiastic interest from multiple communities ranging from condensed matter physics to high energy physics and quantum information sciences. On the one hand, the immobility of fractons may be utilized in quantum computation so as to minimize errors in quantum information processing[6,15–17]. On the other hand, the description of the coupling between fracton and gauge fields requires tensor gauge field theories, a paradigm beyond the conventional vector gauge field theories[18–20]. It is thus expected that fracton phase of matter will offer physicists a rich playground to explore new physics beyond traditional paradigms.

Whereas fracton phase of matter has been explored in a broad range of systems, a recent work has pointed out an elegant scheme to realize such matter using an unconventional bosonic model[21,22]. Consider bosons in a tilted lattice, the energy mismatch between the nearest neighbor sites suppresses the tunnelings of single particles and thus delivers fracton physics in a natural manner. A dipole formed by a particle-hole pair, however, can move in the lattice, due to interaction-induced correlated tunnelings. In one dimension, the system is described by the so-called dipolar Bose-Hubbard model(DBHM),

$$H_2 = -\sum_i \left( t_2 b_i^2 b_{i-1}^\dagger b_{i+1}^\dagger + \text{h.c.} \right) + \frac{U}{2}\sum_i n_i(n_i - 1) \tag{1}$$

where $b_i^\dagger (b_i)$ is the bosonic creation (annihilation) operator at site $i$. $U$ characterizes the usual onsite interaction. The first term in the above equation describes correlated hopping, where two particles initially occupying the same lattice site simultaneously tunnel toward opposite directions. Alternatively, $b_i^\dagger b_{i-1}$ may be regarded as a creation operator for a dipole, and the kinetic energy in Eq. (1) corresponds to the tunneling of a dipole. As shown in Fig. 1a, a notable feature is that a dipole could only move in the direction parallel to itself and is thus confined in a line. Such a dipole is referred to as a lineon. Eq. (1) was also referred as to the constraint Bose-Hubbard model that supports fractonic Luttinger liquids[23]. Constrained spin chains were also studied in the

[1]Department of Physics and Astronomy, Purdue University, West Lafayette, IN 47907, USA. [2]Purdue Quantum Science and Engineering Institute, Purdue University, West Lafayette, IN 47907, USA. ✉e-mail: zhou753@purdue.edu

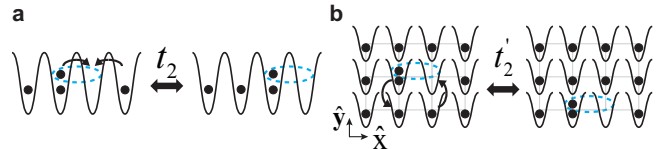

**Fig. 1 | Correlated tunnelings in DBHM. a** The correlated tunneling of two particles allows a dipole to move in the direction parallel to its dipole moment. **b** The ring-exchange interaction allows a dipole to move in the direction perpendicular to its dipole moment.

literature[24]. Similar composite operators of particle-hole pairs were considered in fermionic systems[25]. It is worth pointing out that when the tilting potential between the nearest neighbor sites matches the interaction strength, interaction-assisted single-particle tunneling will take place and lead to very rich phenomena[26]. In our paper, we consider an off-resonant tilted lattice such that tunnelings of single particles are not relevant and Eq. (1) serves as the effective Hamiltonian.

The generalization of Eq. (1) to higher dimensions is straightforward. For instance, in two dimensions, as shown in Fig. 1b, a dipole may be able to tunnel in the transverse direction if we consider

$$H'_2 = -\sum_{\mathbf{i}} \left( t'_2 b^{\dagger}_{\mathbf{i}+\hat{\mathbf{x}}} b^{\dagger}_{\mathbf{i}+\hat{\mathbf{y}}} b_{\mathbf{i}} b_{\mathbf{i}+\hat{\mathbf{x}}+\hat{\mathbf{y}}} + \text{h.c.} \right) + \frac{U}{2}\sum_{\mathbf{i}} n_{\mathbf{i}}(n_{\mathbf{i}} - 1), \quad (2)$$

where $\mathbf{i} = (i_x, i_y)$ is the lattice index, and $\hat{\mathbf{x}}$ and $\hat{\mathbf{y}}$ are the unit vector in the $x$ and the $y$ directions, respectively. The first term is referred to as the ring-exchange interaction. $H'_2$ ensures that a dipole moves in the direction perpendicular to itself. Such a dipole is called a planon. We can certainly include both the ring-exchange interaction and the correlated tunneling as that in $H_2$ so as to allow a dipole to move along all directions.

Unlike BHM that has a global $U(1)$ symmetry, DBHM in Eq. (1) has an additional dipole $U(1)$ symmetry, leading to the dipole conservation law[27]. This can be seen from the fact that $H_2$ in Eq. (1) remains unchanged after adding a phase to the bosonic operator, $b_j \rightarrow b_j e^{i\alpha_j}$ ($b^{\dagger}_j \rightarrow b^{\dagger}_j e^{-i\alpha_j}$), where $\alpha_j = j\varphi$ changes linearly as a function of the lattice index $j$ and $\varphi$ is a constant. Similarly, $H'_2$ in Eq. (2) is invariant if $b_{\mathbf{j}} \rightarrow b_{\mathbf{j}} e^{i\alpha_{\mathbf{j}}}$ and $\alpha_{\mathbf{j}} = j_x\varphi_x + j_y\varphi_y$, where $\varphi_x$ and $\varphi_y$ are constant. The dipole $U(1)$ symmetry of DBHM provides much richer physics than BHM. It has been shown that DBHMs host some intriguing phases as the ground states in different parameter regimes[21,22]. The spontaneous breaking of the dipole $U(1)$ phase leads to a new phase of dipole condensate.

In one phase, $\langle b_{\mathbf{i}} \rangle \neq 0$, $\langle b^{\dagger}_{\mathbf{i}} b_{\mathbf{i}+\hat{\mathbf{x}}} \rangle \neq 0$, and $\langle b^{\dagger}_{\mathbf{i}} b_{\mathbf{i}+\hat{\mathbf{y}}} \rangle \neq 0$. In other words, both the order parameters of single-particle condensates and dipole condensates are finite. The effective Lagrangian then includes only the quartic term of the phase of the single-particle condensate. Whereas it was previously proposed that such Lagrangian can be accessed using synthetic gauge fields and other methods so as to explore the quantum Lifshitz model[13,28–32], DBHMs allow physicists to study it in a broader parameter regime.

The other phase, the dipole condensate defined by $\langle b_{\mathbf{i}} \rangle = 0$, $\langle b^{\dagger}_{\mathbf{i}} b_{\mathbf{i}+\hat{\mathbf{x}}} \rangle \neq 0$, and $\langle b^{\dagger}_{\mathbf{i}} b_{\mathbf{i}+\hat{\mathbf{y}}} \rangle \neq 0$, is of particular interest. Since the realization of a Bose-Einstein condensate in laboratories[33,34], physicists have been continuously exploring unconventional condensates where single particles do not condense but two or more particles first form a pair or a cluster, and then these pairs or clusters condense[35–38]. A variety of schemes have been implemented, such as internal degrees of freedom or multiple band effects. However, because of the intrinsic bosonic statistics, single particles naturally form a condensate at sufficiently low temperatures in generic bosonic systems. Experimental efforts in the past many years have proven the difficulty of accessing unconventional condensates in practice.

The theoretical results of DBHMs suggest a new routine to create dipole condensates in laboratories, for instance, using ultracold atoms in optical lattices. The success of such efforts will provide physicists with many exciting opportunities to explore exotic quantum many-body phenomena. For instance, the condensation of dipoles may gap the tensor gauge fields, as a counterpart of the Higgs mechanism gapping the vector gauge fields[19,20]. Based on recent advancements in ultracold atoms experiments[39–42], it is promising that some exotic phenomena of fractons, dipoles, and dipole condensation, as well as their couplings with gauge fields, may be accessed in laboratories in the near future.

The study of DBHMs imposes some fundamentally important questions. First, whether experimentalists must access the ground state of the DBHMs in Eq. (1) and Eq. (2) so as to create a dipole condensate? If the condensation of dipoles is a more generic phenomenon other than a specific result unique to some particular models, physicists will have a lot more opportunities to explore the fracton phase of matter. Second, in addition to thermal equilibrium, whether a dipole condensate may be created using non-equilibrium quantum dynamics? Studies in the past few decades have shown that non-equilibrium quantum dynamics provide physicists with a much more efficient scheme to access many interesting quantum states than adiabatic processes and other approaches at thermal equilibrium[43]. It is desirable to work out some non-equilibrium quantum processes that may deliver dipole condensates as the desired target states. Third, once a dipole condensate is created, what new macroscopic quantum phenomena may be accessible in experiments? Last but not least, in addition to dipole condensates, how to access a generic multipolar condensate and the resultant macroscopic quantum phenomena?

We answer all the above questions in this work. First, we discuss the general principle for a dipole condensate to arise. Some examples of dipole condensates as familiar normal states of bosons are provided. We show that a self-proximity effect makes dipole condensates much more prevalent than one may naively expect. Secondly, we present schemes to access dipole condensates using non-equilibrium quantum dynamics. Thirdly, we focus on a new macroscopic quantum phenomenon, the dipolar Josephson effect, where the supercurrent of dipoles emerges after the dipole tunnelings act on dipole condensates with different phases. The generalization to a hierarchy of multipolar condensates is also discussed. Discussions about experimental realizations of dipole condensates, dipolar Josephson effects, and multipolar condensates are shown at last.

## Results

### General principles of dipole condensates

A conventional condensate is characterized by a finite single-particle order parameter, $\langle b_{\mathbf{i}} \rangle \neq 0$. As such, the reduced one-body density matrix $\rho_1(\mathbf{i}, \mathbf{j}) \equiv \langle b^{\dagger}_{\mathbf{i}} b_{\mathbf{j}} \rangle$ approaches a finite constant when $|\mathbf{i} - \mathbf{j}| \rightarrow \infty$, defining the off-diagonal long-range order. When the single-particle condensate vanishes, $\langle b_{\mathbf{i}} \rangle = 0$, and $\langle b^{\dagger}_{\mathbf{i}} b_{\mathbf{j}} \rangle \rightarrow 0$ in the large $|\mathbf{i} - \mathbf{j}|$ limit, signifying the vanishing off-diagonal long-range order due to either the interaction effect or thermal fluctuations. Under this situation, the state is often considered as a normal phase of bosons. Figure 2(a) shows a schematic of the phase diagram of BHM. The accurate phase boundary could be obtained from advanced numerics[44,45]. In the normal phase, short-range correlations may still exist and $\langle b^{\dagger}_{\mathbf{i}} b_{\mathbf{j}} \rangle$ remains finite for a finite $|\mathbf{i} - \mathbf{j}|$, characterized by a finite correlation length $l$, as shown in Fig. 2b. In some extreme cases, $\langle b^{\dagger}_{\mathbf{i}} b_{\mathbf{j}} \rangle = n_0 \delta_{\mathbf{i}-\mathbf{j}}$, where $n_0$ is the particle number per site, short-range correlations also vanish. This can be accessed in the strong-interaction limit at zero temperature, where the ground state becomes a Mott insulator,

$$|\text{MI}\rangle = \prod_{\mathbf{i}} \frac{b^{\dagger n_0}_{\mathbf{i}}}{\sqrt{n_0!}} |0\rangle. \quad (3)$$

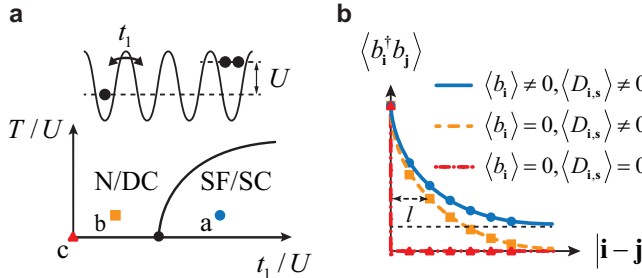

**a**

**b**

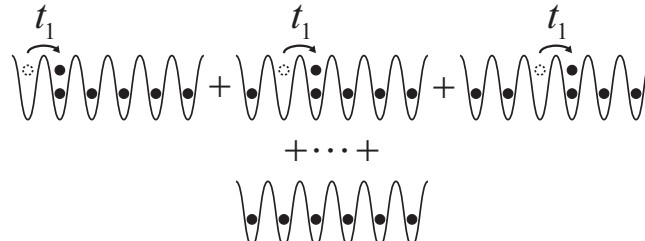

**Fig. 2 | Dipole condensates in BHM. a** The phase diagram of Bose-Hubbard model. SF/SC denotes the single-particle condensate that may also be referred to as the superfluid phase in 3D. N/DC denotes the normal phase that is also a dipole condensate. **b** Behaviors of the reduced one-body density matrix when the parameters are chosen to be three points in (**a**). The correlation length $l$ vanishes in in the limit of $t_1/U{\to}0$ and $T/U{\to}\infty$.

**Fig. 3 | The ground state of BHM in the strong-interaction limit.** In the limit $t_1/U \ll 1$, a superposition of Mott insulator and particle-hole pair excitations accounts for the vanishing off-diagonal long-range order and finite short-range correlations.

Alternatively, the correlation can be suppressed by thermal fluctuations. At the infinite temperature, the correlation length vanishes, and $\langle b_{\mathbf{i}}^\dagger b_{\mathbf{j}} \rangle = n_0 \delta_{\mathbf{i}-\mathbf{j}}$.

A dipole condensate is characterized by a finite two-body order parameter $\langle D_{\mathbf{i,s}} \rangle \neq 0$, where $D_{\mathbf{i,s}} \equiv b_{\mathbf{i}}^\dagger b_{\mathbf{i+s}}$ and $\mathbf{s} \neq 0$. The order parameter, $\langle b_{\mathbf{i}}^\dagger b_{\mathbf{i+\hat{x}}} \rangle$, considered in DBHMs is a special case where the separation between the particle and hole is one lattice spacing. More generically, $D_{\mathbf{i,s}}$ is the creation operator for a dipole, whose center of mass is located at $\mathbf{i} + \mathbf{s}/2$, and $\mathbf{s}$ is regarded as the relative coordinate of the particle-hole pair. In other words, the reduced one-body density matrix captures the order parameter of the dipole condensate. As such, a conventional normal phase of bosons with a finite correlation length $l$ readily acquires a finite dipole order parameter $\langle D_{\mathbf{i,s}} \rangle$ for a finite $\mathbf{s}$, and thus is a dipole condensate. The only exception is the extreme case where the correlation length $l$ vanishes and $\langle D_{\mathbf{i,s}} \rangle$ becomes zero for any finite $\mathbf{s}$. For any finite $l$, the particle and the hole are confined, forming a dipole condensate. In the opposite limit, where the correlation length $l$ becomes divergent, the particle and the hole become deconfined. This is accompanied by the rise of the single-particle condensate and the single-particle order parameter $\langle b_{\mathbf{i}} \rangle$ becomes finite.

The prevalence of a dipole condensate can also be understood from a different perspective. The order parameter of a dipole condensate can be regarded as the Fourier transform of the momentum distribution,

$$\langle D_{\mathbf{i,s}} \rangle = \frac{1}{L} \sum_{\mathbf{k}} e^{i\mathbf{k} \cdot (\mathbf{R_{i+s}} - \mathbf{R_i})} n_{\mathbf{k}}, \qquad (4)$$

where $n_{\mathbf{k}} = \langle a_{\mathbf{k}}^\dagger a_{\mathbf{k}} \rangle$ is the momentum distribution, $a_{\mathbf{k}}^\dagger (a_{\mathbf{k}})$ is the creation (annihilation) operator in the momentum space, $L$ is the number of lattice sites, and $\mathbf{R_{i+s}}$ is the coordinate of site $\mathbf{i} + \mathbf{s}$. We have considered homogeneous systems such that $\langle a_{\mathbf{k}}^\dagger a_{\mathbf{q}} \rangle = 0$ if $\mathbf{k} \neq \mathbf{q}$. If a single-particle condensate exists, $n_{\mathbf{k}} \cdot N_0 \delta_{\mathbf{k}}$, where $N_0$ is of the order of the total particle number $N$. The above equation immediately tells us that $\langle b_{\mathbf{i}}^\dagger b_{\mathbf{i+s}} \rangle$ is of the order of $n_0 = N/L$, signifying a dipole condensate. This is what one naturally expects, as a finite single-particle order parameter $\langle b_{\mathbf{i}} \rangle$ automatically leads to a finite two-body order parameter $\langle b_{\mathbf{i}}^\dagger b_{\mathbf{j}} \rangle \sim \langle b_{\mathbf{i}}^\dagger \rangle \langle b_{\mathbf{j}} \rangle$. But a single-particle condensate is only a sufficient, not a necessary, condition for a dipole condensate. Even when the single-particle condensate vanishes and $n_{\mathbf{k}}$ may have a broad distribution in the momentum space, the right-hand side of Eq. (4) may still lead to a constructive interference such that $\langle b_{\mathbf{i}}^\dagger b_{\mathbf{i+s}} \rangle$ is of the order of $n_0$. For instance, $n_{\mathbf{k}}$ is a constant in a "Fermi" sea. We will discuss such examples in the non-equilibrium preparation section.

We emphasize that the existence of a dipole condensate does not require the dipole conservation law in the system. If the dipole is conserved, for instance, in the DBHMs, the charge is strictly immobile, as the movement of a charge inevitably creates an extra dipole. Even in the absence of the dipole conservation law, the motion of a charge may still be constrained. In a generic normal phase of bosons, either thermal or quantum fluctuations destroy the off-diagonal long-range order, and the correlation length $l$ becomes finite. A dipole condensate could therefore be defined. It is also worth pointing out that, though the theoretical definition of a fraction phase of matter relies on the dipole conservation law, the realization of such a phase in practice, may involve microscopic physics that breaks the dipole conservation law. For instance, a tilted lattice suppresses the single-particle tunneling such that the dominant pair tunneling creates the DBHM as an effective theory at low energies. Nevertheless, single-particle tunneling still exists no matter how strong the tilting is, though it does not affect low-energy physics. In this sense, the dipole condensate is a fracton phase of matter, regardless of whether it arises from DBHM or BHM. As previously explained, despite the absence of the dipole conservation law in BHM, single-particle tunneling is suppressed at large distances (or low energies) in the normal phase.

## Dipole condensates in BHM

We first consider a conventional Bose-Hubbard model(BHM),

$$H_1 = - \sum_{\langle \mathbf{i,j} \rangle} \left( t_1 b_{\mathbf{i}}^\dagger b_{\mathbf{j}} + \text{h.c.} \right) + \frac{U}{2} \sum_{\mathbf{i}} n_{\mathbf{i}}(n_{\mathbf{i}} - 1), \qquad (5)$$

where $\langle \mathbf{i, j} \rangle$ denotes the nearest neighbor sites. At zero temperature, when $t_1/U$ is larger than a critical value $(t_1/U)_c$, the ground state is a condensate, $\langle b_{\mathbf{i}} \rangle \neq 0$. When $t_1/U < (t_1/U)_c$, the off-diagonal long-range order and the single-particle condensate vanish. The ground state becomes a Mott insulator[46,47]. Nevertheless, short-range correlations exist provided that $t_1/U$ is finite. For instance, when $t_1/U \ll 1$, the ground state can be well approximated by

$$|\Psi_1\rangle \approx |\,\text{MI}\,\rangle + \frac{t_1}{U} \sum_{\langle \mathbf{i,j} \rangle} \left( \hat{b}_{\mathbf{i}}^\dagger \hat{b}_{\mathbf{j}} + \text{h.c.} \right) |\,\text{MI}\,\rangle, \qquad (6)$$

as shown in Fig. 3.

Eq. (6) is obtained by regarding the tunneling term $t_1 b_{\mathbf{i}}^\dagger b_{\mathbf{j}} + \text{h.c.}$ as the perturbation to the interaction term with only the first order in $t_1/U$ kept. This approach faithfully captures short-range correlations in the small $t_1/U$ limit, unlike the Gutzwiller's ansatz that leads to the immediate disappearance of both long-range and short-range correlations once $t_1/U$ is smaller than the critical value $(t_1/U)_c$ separating the superfluid phase and Mott insulator. In reality, only when $t_1/U = 0$, the ground state becomes a perfect Mott insulator with vanishing short-range correlation. For any finite $t_1/U$, it is known and has been observed in experiments that Mott insulators do exhibit short-range correlations, which are captured by Eq. (6) in the limit $t_1/U{\to}0$[48]. Using Eq. (6),

it is straightforward to obtain

$$\langle D_{\mathbf{i},\hat{\mathbf{x}}}\rangle = \langle D_{\mathbf{i},\hat{\mathbf{y}}}\rangle = \frac{t_1}{U} 2n_0(n_0+1), \tag{7}$$

from which we conclude that the Mott insulator with short-range correlation is readily a dipole condensate. The reduced one-body density matrix of the dipole, $\langle D_{\mathbf{i},\mathbf{s}}^\dagger D_{\mathbf{j},\mathbf{r}}\rangle$, approaches a constant in the large $|\mathbf{i}-\mathbf{j}|$ limit. Here, $\mathbf{s}$ and $\mathbf{r}$ denote $\hat{\mathbf{x}}$ or $\hat{\mathbf{y}}$.

More generically, we could evaluate the reduced single-particle density matrix $\rho_1$,

$$\rho_1(\mathbf{i,j}) = \begin{cases} n_0, & |\mathbf{i}-\mathbf{j}|=0, \\ \frac{t_1}{U} 2n_0(n_0+1), & |\mathbf{i}-\mathbf{j}|=1. \end{cases} \tag{8}$$

The momentum distribution is simply the Fourier transform of $\rho_1$,

$$n(\mathbf{k}) = n_0 + 4\frac{t_1}{U}n_0(n_0+1)(\cos k_x d + \cos k_y d), \tag{9}$$

where $d$ is the lattice constant. Substituting Eq. (9) to Eq. (4), we see that the first term, $n_0$, does not contribute to the right-hand side of Eq. (4), as $\sum_{\mathbf{k}} e^{i\mathbf{k}\cdot(\mathbf{R}_{\mathbf{i}+\mathbf{s}}-\mathbf{R}_{\mathbf{i}})}=0$ for any finite $\mathbf{s}$. In contrast, the second term on the right-hand side of Eq. (9) leads to a constructive interference on the right-hand side of Eq. (4). As such, $\langle D_{\mathbf{i},\hat{\mathbf{x}}}\rangle$ and $\langle D_{\mathbf{i},\hat{\mathbf{y}}}\rangle$ become finite despite that the single-particle condensate is absent.

When $t_1/U$ increases, Eq. (9) is no longer a good approximation. Nevertheless, the same conclusion about the dipole condensate applies. The only quantitative difference is that $\langle D_{\mathbf{i},\mathbf{s}}\rangle$ becomes finite for a generic $\mathbf{s}$. The width of the relative wavefunction of the dipole increases while the dipole condensate remains finite. To be more explicit, the reduced one-body density matrix of the dipole, $\langle D_{\mathbf{i},\mathbf{s}}^\dagger D_{\mathbf{j},\mathbf{r}}\rangle$, remains finite when $|\mathbf{i}-\mathbf{j}|\to\infty$ for generic $\mathbf{s}$ and $\mathbf{r}$. When $t_1/U = (t_1/U)_c$, dipoles become deconfined and a single-particle condensate arises. In the opposite limit, $t_1/U = 0$ and the dipole condensate vanishes.

Another example is the normal phase at finite temperatures. For instance, considering the high-temperature regime, $T \gg t_1 \gg U$, the momentum distribution can be written as

$$n_{\mathbf{k}} = e^{\beta(\mu-\epsilon_{\mathbf{k}})}, \tag{10}$$

where $\mu$ is the chemical potential and $\epsilon_{\mathbf{k}}$ is the single-particle energy. In the dilute limit, $\epsilon_{\mathbf{k}} = \hbar^2 k^2/(2m^*)$, where $k$ is the magnitude of $\mathbf{k}$ and $m^*$ is the effective mass at the band bottom. Substituting Eq. (10) to Eq. (4), we obtain

$$\langle D_{\mathbf{i},\mathbf{s}}\rangle = n_0 \frac{\sum_{\mathbf{k}} e^{-i\mathbf{k}\cdot(\mathbf{R}_{\mathbf{i}+\mathbf{s}}-\mathbf{R}_{\mathbf{i}})} e^{-\beta\epsilon_{\mathbf{k}}}}{\sum_{\mathbf{k}} e^{-\beta\epsilon_{\mathbf{k}}}}. \tag{11}$$

The numerator in Eq. (11) leads to a constructive interference such that $\langle D_{\mathbf{i},\mathbf{s}}\rangle$ become finite for any finite $\mathbf{s}$. For instance, we may consider a one-dimensional gas at the high-temperature regime $T \gg T_{1D}$, where $T_{1D} \equiv 2\pi\hbar^2 n_0^2/m^* d^2$. It is straightforward to obtain the momentum distribution $n_k = \sqrt{T_{1D}/T}e^{-\epsilon_k/T}$, and therefore $\langle D_{\mathbf{i},\hat{\mathbf{x}}}\rangle = n_0 e^{-m^* T d^2/2\hbar^2}$. We thus conclude that the normal phase at finite temperatures is readily a dipole condensate as well. Only in the limit of the infinite temperature, $\langle D_{\mathbf{i},\hat{\mathbf{x}}}\rangle$ is suppressed down to zero and the dipole condensate vanishes.

## Self-proximity effects

The above results of dipole condensate can be understood as a self-proximity effect. The proximity effect has played a vital role in the study of superconductivity[49]. When a superconductor is placed in contact with a normal metal, Cooper pairs tunnel through the interface. Such a tunneling process acts as a linear term of the creation or

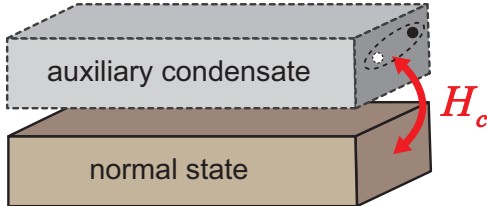

**Fig. 4 | The self-proximity effect.** A normal state is coupled to an auxiliary condensate by the tunnelings of particle-hole pairs that automatically induce dipole condensation in the normal state. Here, no external condensate is required in reality and the single-particle kinetic energy plays the role of the auxiliary condensate.

annihilation of Cooper pairs and thus automatically induces superconductivity in the normal state.

Here, we can think about BHM in a similar means. The first term in Eq. (5), i.e., the kinetic energy of single particles, is readily a linear term of the creation (or annihilation) operator of dipoles. In the mean-field approach, it corresponds to a linear term of the order parameter of the dipole. As such, no matter how weak it is, a finite dipole order parameter must be produced.

To be more explicit, we may imagine that there is an auxiliary condensate coupled to the system of interest, as shown by Fig. 4. The Hamiltonian of the composite system is written as $H = \tilde{H}_1 + H_a + H_c$, where

$$\tilde{H}_1 = \frac{U}{2}\sum_{\mathbf{i}} n_{\mathbf{i}}(n_{\mathbf{i}}-1), \tag{12}$$

$$H_a = -\sum_{\langle \mathbf{i,j}\rangle}\left(t_a c_{\mathbf{i}}^\dagger c_{\mathbf{j}} + \text{h.c.}\right), \tag{13}$$

and

$$H_c = -\sum_{\langle \mathbf{i,j}\rangle}\left(\Omega_{\mathbf{i,j}} b_{\mathbf{i}}^\dagger b_{\mathbf{j}} c_{\mathbf{j}}^\dagger c_{\mathbf{i}} + \text{h.c.}\right). \tag{14}$$

Because of the absence of kinetic energy, the ground state of $\tilde{H}_1$ is a Mott insulator with no single-particle condensation or dipole condensation. At finite temperatures, both condensates are still absent.

In contrast, the lack of interactions in $H_a$ tells us that the ground state of the auxiliary system is a condensate $\langle c_{\mathbf{i}}\rangle = \phi$. $H_c$ denotes the hopping of particle-hole pairs between the bosonic system of interest and the auxiliary condensate, as analogous to the particle-particle pair coupling in conventional proximity effects of superconductors. The Mott insulator and the auxiliary condensate play the roles of the normal metal and the superconductor in the ordinary proximity effect.

The condensation of the auxiliary system allows us to replace $c_{\mathbf{j}}^\dagger c_{\mathbf{i}}$ by its expectation value and $H_c$ becomes

$$H_{c,\text{mf}} = -\sum_{\langle \mathbf{i,j}\rangle}\left(\Omega_{\mathbf{i,j}}|\phi|^2 b_{\mathbf{i}}^\dagger b_{\mathbf{j}} + \text{h.c.}\right). \tag{15}$$

This is precisely a counterpart of the induced Cooper pairing in the conventional proximity of superconductors, where $|\phi|^2$ plays the role of the order parameter of the superconductor. As such, an auxiliary condensate automatically induces a dipole condensate, $\langle b_{\mathbf{i}}^\dagger b_{\mathbf{j}}\rangle \neq 0$, regardless of how strong the finite interaction $U$ is. The same conclusion applies to finite temperatures where the Mott insulator is replaced by a generic normal state.

As a comparison, we consider a different coupling between the auxiliary condensate and the system of interest,

$$H'_c = - \sum_i \left( \Omega'_i b^\dagger_i c_i + \text{h.c.} \right).$$ (16)

Now the Mott insulator and the auxiliary condensate are coupled by the single-particle tunneling. A finite $\langle c_i \rangle = \phi'$ gives rise to the mean-field Hamiltonian,

$$H'_{c,\text{mf}} = - \sum_i \left( \Omega'_i \phi' b^\dagger_i + \text{h.c.} \right).$$ (17)

$\Omega'_i \phi'$ and its hermitian conjugate serve as the source and the drain of single particles, respectively. A finite $\phi'$ thus produces a finite $\langle b^\dagger_i \rangle$, regardless of how large $U$ is. This time, a single-particle condensate is induced. We define $H_0 = H'_{c,\text{mf}} + \tilde{H}_1$, which can be written as

$$H_0 = - \sum_i \left( t_0 b^\dagger_i + \text{h.c.} \right) + \frac{U}{2} \sum_i n_i(n_i - 1).$$ (18)

We have considered a site-independent $\Omega'_i \equiv \Omega'$ to simplify notations, and $t_0 = \Omega' \phi'$. The ground state of $H_0$ is always a single-particle condensate even in the large $U$ limit, as analogous to the dipole condensate as the ground state of $H_1$ in the large $U$ limit.

Whereas the above discussions resemble that of the conventional proximity effect, we must emphasize an intrinsic difference. Here, no other quantum systems are required in reality and the auxiliary condensate is introduced only for the purpose of understanding the underlying physics. The proximity effect is a self-induced one. The kinetic energy of single particles in $H_1$, i.e., $b^\dagger_i b_j$, readily provides the source of the dipole condensate, similar to $b^\dagger_i$ as the single-particle source in $H_0$. We thus name this effect a self-proximity effect. Such an effect also exists in a more generic situation when we consider an arbitrary multipolar condensate later.

As previously explained, DBHM is characterized by the dipole $U(1)$ symmetry and thus has the dipole conservation law. Spontaneous symmetry breaking of the dipole $U(1)$ symmetry provides DBHM with a rich phase diagram including a dipole condensate as the ground state. In contrast, the dipole conservation law is absent in BHM. A dipole condensate arises from a completely different microscopic mechanism, the self-proximity effect. Nevertheless, the off-diagonal long-range orders of these two cases are the same, for instance, $\langle b_i \rangle = 0$, $\langle b^\dagger_i b_{i+\hat{x}} \rangle \neq 0$, and $\langle b^\dagger_i b_{i+\hat{y}} \rangle \neq 0$. An important difference between these two cases is that the phase of a dipole condensate in BHM is fixed by the single-particle tunneling, unlike the ground state of DBHM where the spontaneous symmetry breaking allows an arbitrary phase of the dipole condensate. This is similar to an ordinary proximity effect, where the phase of the induced condensate or superfluid is determined by external conditions, not by spontaneous symmetry breaking. Here, controlling the phase of the single-particle tunneling provides experimentalists with a unique means to tune and twist the phase of a dipole condensate. Turning on correlated hoppings then gives rise to a dipolar Josephson effect.

## Non-equilibrium preparation of dipole condensates

Eq. (4) tells us that a broad range of $n_k$ may produce a dipole condensate. In particular, we could consider $n_k$ unattainable at equilibrium. In fact, non-equilibrium dynamics have been widely used as an efficient tool to prepare novel quantum states in laboratories. Here, we consider a bosonic metal, whose

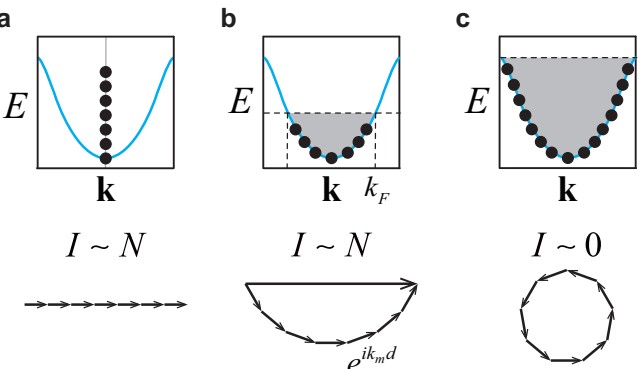

**Fig. 5 | The momentum distribution and the dipole order parameter.** These three panels denote the results of $I \equiv \sum_k e^{i k \cdot (R_{i+s} - R_i)} n_k$ for a single-particle condensate (**a**), a bosonic metal (**b**), and a band insulator (**c**).

momentum distribution resembles that of non-interacting fermions,

$$n_k = \begin{cases} n_0, & \epsilon_k < E_F, \\ 0, & \epsilon_k > E_F, \end{cases}$$ (19)

where $E_F$ is the "Fermi" energy. We have previously discussed that using the momentum distribution of a single-particle condensate, $n_k - N_0 \delta_k$, $I \equiv \sum_k e^{i k \cdot (R_{i+s} - R_i)} n_k$ is of the order of $N$ and thus leads to a finite order parameter of dipole. Here, using $n_k$ of a bosonic metal, $I$ is also of the order of $N$, and $\langle D_{i,s} \rangle$ is of the order of $N/L$. A pictorial description of such constructive interference is shown in Fig. 5. A bosonic metal is thus a dipole condensate.

It is worth mentioning that, to deliver a dipole condensate, the occupied states do not need to be the ones with the lowest single-particle energies. For instance, the bosonic cloud can be shifted by a constant $k_0$ in the momentum space. The dipole order parameter becomes a function of $k_0$,

$$\langle D_{i,s} \rangle (k_0) = e^{i k_0 \cdot (R_{i+s} - R_i)} \langle D_{i,s} \rangle (0).$$ (20)

In other words, the dipole condensate acquires a **s**-dependent phase. $n_k$ could also be a more generic function in addition to a step function. The previously discussed finite temperature result at thermal equilibrium is such an example.

When $E_F$ continues to increase, all states in the Brillouin zone (BZ) are eventually occupied, and the bosonic metal turns into a bosonic band insulator,

$$|\text{BI}\rangle = \prod_{k \in BZ} \frac{a^{\dagger n_0}_k}{\sqrt{n_0!}} |0\rangle.$$ (21)

Previous studies have considered such a state in one dimension with $n_0 = 1$ in the context of a $N$-port Hong-Ou-Mandel interferometer in quantum optics[50,51]. It has been shown that certain output events are strictly prevented. In our language, this corresponds to dipole conservation. Whereas the real-space representation of $|\text{BI}\rangle$ includes many product states, the dipole moment mod $N$ is conserved, i.e., $(P \bmod N) = 0$, where $P = \sum_i i n_i$ is the dipole moment of a product state with the occupation number at site $i$ denoted by $n_i$. Any state that does not satisfy this conservation law is, therefore, suppressed in the output of the interferometer. Here, Eq. (4) shows that $\langle D_{r,s} \rangle$ becomes zero for any finite **s** and the dipole condensate vanishes. Adding a finite doping creates a bosonic metal and thus a dipole condensate.

**Article** https://doi.org/10.1038/s41467-024-48907-9

## Multiport Hong-Ou-Mandel interferometers

Now we discuss how to create a bosonic metal in laboratories. A recent experiment has implemented an adiabatic process to transfer the occupancy in the real space to the momentum space[52]. This method could, in principle, deliver a bosonic metal by populating some lattice sites in real space and then adiabatically converting them to desired momentum states. However, at the final stage, **k** and −**k** are typically degenerate, imposing a challenge to retain adiabaticity. Moreover, it is time-consuming to adopt adiabatic processes in reality and a shortcut to adiabaticity is often appreciated. To this end, we present a preparation method using non-equilibrium quantum dynamics. The idea is to design a Hamiltonian as the generator of the required transform matrix. On a 1D lattice with $L$ sites, the required transform matrix is the Discrete Fourier Transform (DFT) matrix,

$$\mathcal{F} = \frac{1}{\sqrt{L}} \begin{pmatrix} 1 & 1 & \cdots & 1 & 1 \\ 1 & \omega & \cdots & \omega^{L-2} & \omega^{L-1} \\ \vdots & \vdots & \cdots & \vdots & \vdots \\ 1 & \omega^{L-1} & \cdots & \omega^{(L-2)(L-1)} & \omega^{(L-1)(L-1)} \end{pmatrix} \quad (22)$$

where $\omega = e^{2\pi i/L} = \cos(2\pi/L) + i\sin(2\pi/L)$. Acting $\mathcal{F}$ on creation operators in the real space provides operators in the momentum space, namely

$$\mathcal{F} \begin{pmatrix} b_1^\dagger \\ b_2^\dagger \\ \vdots \\ b_L^\dagger \end{pmatrix} = \begin{pmatrix} a_{k_1}^\dagger \\ a_{k_2}^\dagger \\ \vdots \\ a_{k_L}^\dagger \end{pmatrix}, \quad (23)$$

where $k_i = 2\pi(i-1)/L$ and $i = 1,2,...,L$.

If all lattice sites are initially occupied by one particle, after applying $\mathcal{F}$, we obtain a band insulator. Similarly, this method could deliver an arbitrary distribution of particles in the momentum space. For instance, to obtain a bosonic metal, experimentalists just need to fill the lattice sites corresponding to the momentum states within the "Fermi" sea.

To realize a DFT in experiments, we implement an interesting property of the DFT matrix. If we define

$$\mathcal{H} = \frac{1}{2}(1 - (1+i)\mathcal{F} + \mathcal{F}^2 - (1-i)\mathcal{F}^3), \quad (24)$$

it has been shown that[53]

$$\mathcal{F} = e^{i\mathcal{H}\pi/2}. \quad (25)$$

If we regard $\mathcal{H}$ as a generator of time translation and define a Hamiltonian

$$H = -\begin{pmatrix} b_1^\dagger & b_2^\dagger & \dots & b_L^\dagger \end{pmatrix} \mathcal{H} \begin{pmatrix} b_1 \\ b_2 \\ \vdots \\ b_L \end{pmatrix}, \quad (26)$$

$\mathcal{F}$ is a $\pi/2$ pulse that establishes a one-to-one mapping between lattice sites and the momentum states via $k_j = 2\pi(j-1)/L$, according to $e^{-iH\pi/2} b_j^\dagger e^{iH\pi/2} = a_{k_j}^\dagger$.

If the initial state is a Mott insulator, we obtain

$$e^{-iH\pi/2}|\text{MI}\rangle = |\text{BI}\rangle. \quad (27)$$

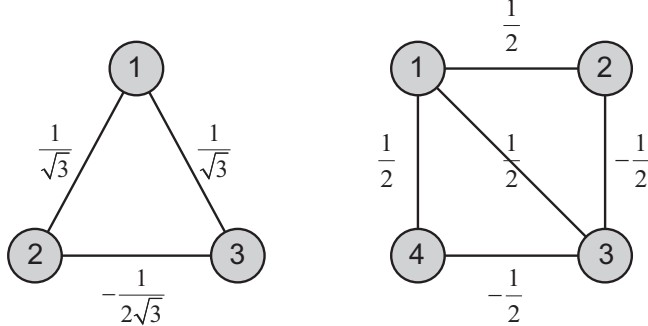

**Fig. 6 | Lattice models producing discrete Fourier transformations.** These two panels show the required Hamiltonians for preparing a bosonic metal in a lattice with $L = 3$ and $L = 4$ sites. The numbers on bonds denote the tunneling strengths between lattice sites.

Alternatively, if the lattice sites are partially filled in the initial state, $|\text{PI}\rangle = \prod_{i=1}^{i_M} b_i^{\dagger n_0} / \sqrt{n_0!}|0\rangle$, where $i_M < L$. $b_i^\dagger$ $(i = 1,2,...i_M)$ is mapped to $a_{k_i}^\dagger$ by this $\pi/2$ pulse, where $|k_i| < k_F$ and $k_F$ is the "Fermi" momentum. $|\text{PI}\rangle$ becomes the desired bosonic metal,

$$e^{-iH\pi/2}|\text{PI}\rangle = |\text{BM}\rangle \equiv \prod_{k<k_F} \frac{a_k^{\dagger n_0}}{\sqrt{n_0!}}|0\rangle. \quad (28)$$

As an example, when $L = 2$, the DFT matrix is simply

$$\mathcal{F} = \frac{1}{\sqrt{2}} \begin{pmatrix} 1 & 1 \\ 1 & -1 \end{pmatrix}. \quad (29)$$

Based on Eq. (26), the Hamiltonian needs to be

$$H = -\begin{pmatrix} b_1^\dagger & b_2^\dagger \end{pmatrix} \begin{pmatrix} 1 - \frac{1}{\sqrt{2}} & -\frac{1}{\sqrt{2}} \\ -\frac{1}{\sqrt{2}} & 1 + \frac{1}{\sqrt{2}} \end{pmatrix} \begin{pmatrix} b_1 \\ b_2 \end{pmatrix}. \quad (30)$$

It is straightforward to verify that $e^{-iH\pi/2} b_1^\dagger e^{iH\pi/2} = (b_1^\dagger + b_2^\dagger)/\sqrt{2}$ and $e^{-iH\pi/2} b_2^\dagger e^{iH\pi/2} = (b_1^\dagger - b_2^\dagger)/\sqrt{2}$. Each of these two lattice sites is uniquely mapped to a state in the momentum space.

The above discussion can be applied to an arbitrary $L$. Here we show the results of Eq. (26) when $L = 3$,

$$H = -\begin{pmatrix} b_1^\dagger & b_2^\dagger & b_3^\dagger \end{pmatrix} \begin{pmatrix} 1 - \frac{1}{\sqrt{3}} & -\frac{1}{\sqrt{3}} & -\frac{1}{\sqrt{3}} \\ -\frac{1}{\sqrt{3}} & 1 + \frac{1}{2\sqrt{3}} & \frac{1}{2\sqrt{3}} \\ -\frac{1}{\sqrt{3}} & \frac{1}{2\sqrt{3}} & 1 + \frac{1}{2\sqrt{3}} \end{pmatrix} \begin{pmatrix} b_1 \\ b_2 \\ b_3 \end{pmatrix}, \quad (31)$$

and when $L = 4$,

$$H = -\begin{pmatrix} b_1^\dagger & b_2^\dagger & b_3^\dagger & b_4^\dagger \end{pmatrix} \begin{pmatrix} \frac{1}{2} & -\frac{1}{2} & -\frac{1}{2} & -\frac{1}{2} \\ -\frac{1}{2} & 1 & \frac{1}{2} & 0 \\ -\frac{1}{2} & \frac{1}{2} & \frac{1}{2} & \frac{1}{2} \\ -\frac{1}{2} & 0 & \frac{1}{2} & 1 \end{pmatrix} \begin{pmatrix} b_1 \\ b_2 \\ b_3 \\ b_4 \end{pmatrix}. \quad (32)$$

The off-diagonal elements of $H$ represent the tunneling strengths among lattice sites, as shown in Fig. 6. If these lattice sites are aligned in a one-dimensional chain, tailored long-range tunnelings are required. Such long-range tunnelings can be created using nanophotonics, cavities, trapped ions, or superconducting circuits[54–58]. Alternatively, movable optical tweezers could be implemented to design tunnelings between any arbitrary pair of lattice sites in the same manner as

creating the graph states[59]. It was also proposed that a sequence of local Hamiltonians could produce Fourier transform[60].

This method can be generalized to higher dimensions. For example, on a $L_x \times L_y$ square lattice, $\mathcal{F}$ is defined as

$$
\mathcal{F}
\begin{pmatrix}
b_{1,1}^\dagger \\
b_{1,2}^\dagger \\
\vdots \\
b_{i,j}^\dagger \\
\vdots
\end{pmatrix}
=
\begin{pmatrix}
a_{k_1,k_1}^\dagger \\
a_{k_1,k_2}^\dagger \\
\vdots \\
a_{k_i,k_j}^\dagger \\
\vdots
\end{pmatrix},
\tag{33}
$$

where $k_{i(j)} = 2\pi(i(j) - 1)/L_{y(x)}$, and $i(j) = 1,2,...,L_{y(x)}$. Since $\mathcal{F}$ describes 2D Fourier transformation, it can be written as a product of two 1D Fourier transformations, namely $\mathcal{F} = \mathcal{F}_y \mathcal{F}_x$. $\mathcal{F}_x$ acts on chains along the $x$ direction, and it is a block diagonal matrix including $L_y$ blocks,

$$
\mathcal{F}_x =
\begin{pmatrix}
\mathcal{D}_x & & & & \\
& \ddots & & & \\
& & \mathcal{D}_x & & \\
& & & \ddots & \\
& & & & \mathcal{D}_x
\end{pmatrix},
\tag{34}
$$

where $\mathcal{D}_x$ is a $L_x \times L_x$ block acting on each chain in the $x$ direction, as shown by Eq. (22) with $L$ replaced by $L_x$. Similarly, $\mathcal{F}_y$ acts on chains along $y$ direction. It can be block diagonalized by rearranging the order of the basis $b_{i,j}^\dagger$ based on the index $j$. Both $\mathcal{F}_x$ and $\mathcal{F}_y$ can be realized using $\pi/2$ pulse, as we previously discussed. As an example, we consider the $L_x = L_y = 2$ case,

$$
\begin{aligned}
\mathcal{F} &= \frac{1}{2}
\begin{pmatrix}
1 & 1 & 1 & 1 \\
1 & -1 & 1 & -1 \\
1 & 1 & -1 & -1 \\
1 & -1 & -1 & 1
\end{pmatrix} \\
&= \frac{1}{\sqrt{2}}
\begin{pmatrix}
1 & 0 & 1 & 0 \\
0 & 1 & 0 & 1 \\
1 & 0 & -1 & 0 \\
0 & 1 & 0 & -1
\end{pmatrix}
\cdot
\frac{1}{\sqrt{2}}
\begin{pmatrix}
1 & 1 & 0 & 0 \\
1 & -1 & 0 & 0 \\
0 & 0 & 1 & 1 \\
0 & 0 & 1 & -1
\end{pmatrix} \\
&\equiv \mathcal{F}_y \mathcal{F}_x.
\end{aligned}
\tag{35}
$$

As $\mathcal{F}_x$ and $\mathcal{F}_y$ commute with each other, we may apply these two $\pi/2$ pulses in an arbitrary order. The required Hamiltonian for each step can be obtained following the previous discussions about one dimension.

## Dipolar Josephson effect

To access macroscopic quantum phenomena, it is crucial to manipulate the phase of a condensate. Here, our results can be utilized to imprint tailored phase patterns to dipole condensates. In BHM as shown in Eq. (5), an additional phase can be added to the tunneling $t_1$, $t_1 \to t_1 e^{i\theta}$, using synthetic gauge fields[61–63]. This amounts to providing the dipole condensate with a phase, $\langle D_{\mathbf{i},\mathbf{i}+\hat{\mathbf{x}}} \rangle \to \langle D_{\mathbf{i},\mathbf{i}+\hat{\mathbf{x}}} \rangle e^{i\theta}$. This phase can be further made position-dependent. For instance, we consider two chains, each of which is described by a BHM, and the tunnelings $t_1$ have different phases. This phase difference could be used to create supercurrents of dipoles and dipolar Josephson effect.

The dipolar Josephson effect is shown in Fig. 7a. Initially, the Hamiltonian is written as

$$
H_i = -\sum_{l,j} \left( t_1 e^{i\theta_l} b_{l,j+1}^\dagger b_{l,j} + \text{h.c.} \right) + \frac{U}{2} \sum_{l,j} n_{l,j}(n_{l,j} - 1),
\tag{36}
$$

where $l = 1,2$ denote the upper and the lower chain, respectively.

As we previously discussed, when $t_1/U < (t_1/U)_c$, the ground state of $H_i$ is a dipole condensate. Here, a critical ingredient is that a phase difference $\Delta\theta \equiv \theta_1 - \theta_2$ between the upper chain and the lower chain has been imprinted to the system. At time $\tau = 0$, we turn on a ring-exchange interaction and change the Hamiltonian to the DBHM,

$$
\begin{aligned}
H_f = &-\sum_j \left( t_2 b_{1,j}^\dagger b_{2,j+1}^\dagger b_{1,j+1} b_{2,j} + \text{h.c.} \right) \\
&+ \frac{U}{2} \sum_{l,j} n_{l,j}(n_{l,j} - 1).
\end{aligned}
\tag{37}
$$

The ground state of $H_i$ is no longer an eigenstate of $H_f$, the dipole tunneling $t_2$ and a finite $\Delta\theta$ jointly create a dipolar Josephson effect.

We define the dipole moment of each chain,

$$
P_1(\tau) = \sum_j j n_{1,j}, \quad P_2(\tau) = \sum_j j n_{2,j}.
\tag{38}
$$

And the total dipole moment is

$$
P(\tau) = \sum_{l,j} j n_{l,j}(\tau).
\tag{39}
$$

As shown in Fig. 7b, $P(\tau)$ is a time-independent constant. This is expected, as the kinetic energy in $H_f$ conserves the total dipole moment. However, both $P_1(\tau)$ and $P_2(\tau)$ are time-dependent. We thus define the supercurrent of dipoles as

$$
J_D(\tau) = \langle \frac{\partial P_1}{\partial \tau} \rangle = - \langle \frac{\partial P_2}{\partial \tau} \rangle.
\tag{40}
$$

Figure 7c shows the result of $J_D(\tau)$. In later times, $J_D(\tau)$ becomes a complex function of time, dependent on the competition between $t_2$ and $U$. When $\tau \to 0$, there exists simple Josephson relation,

$$
J_D = A \sin(\theta_1 - \theta_2),
\tag{41}
$$

where $A$ is a time-independent constant. This can be understood from that the supercurrent of dipoles is written as

$$
\begin{aligned}
J_D &= i \langle [H_f, P_1(\tau)] \rangle \\
&= it_2 \sum_j \left( \langle D_{1,j}^\dagger D_{2,j} \rangle - \langle D_{2,j}^\dagger D_{1,j} \rangle \right),
\end{aligned}
\tag{42}
$$

where $D_{l,j} = b_{l,j}^\dagger b_{l,j+1}$. When $\tau \to 0$, the average is taken at the initial state, and $\langle D_{1,j}^\dagger D_{2,j} \rangle = \langle D_{1,j}^\dagger \rangle \langle D_{2,j} \rangle \sim e^{i(\theta_1 - \theta_2)}$. $J_D$ is thus given by Eq. (41). The constant $A$ can be derived in certain extreme cases,

$$
A =
\begin{cases}
8t_2 \left( \frac{t_1}{U} \right)^2 n_0^2 (n_0 + 1)^2 (L - 1), & \frac{t_1}{U} \ll 1, \\
t_2 n_0^2 \left[ \cos\left( \frac{2\pi}{L+1} \right) + 2 \right] \frac{L^2}{L+1}, & \frac{t_1}{U} \gg 1.
\end{cases}
\tag{43}
$$

We have made use of the fact that when $t_1/U \ll 1$, the ground state of BHM is given by Eq. (6). In the opposite limit $t_1/U \gg 1$, the ground state can be approximated by a single-particle condensate where all particles in each chain are condensed at the zero momentum state. These analytical results agree with numerical simulations well, as shown in Fig. 7d, e. Here the tunneling of planons may happen on

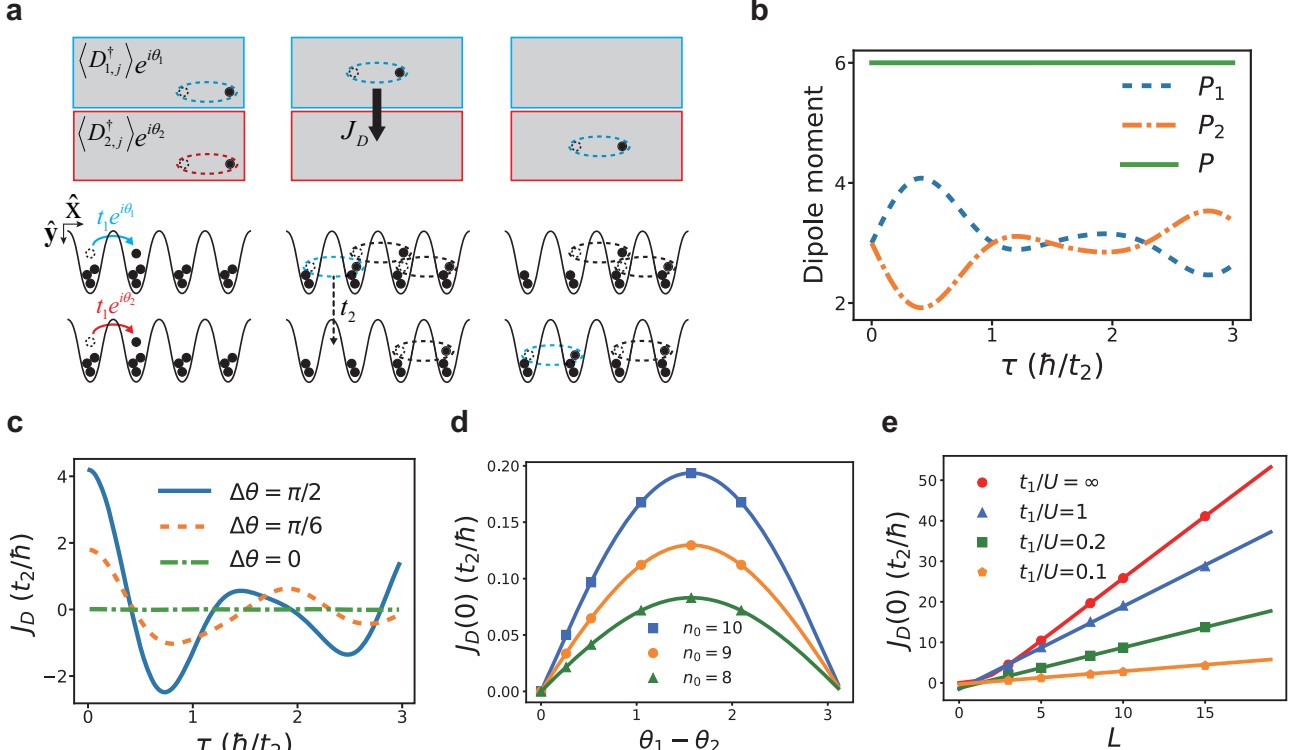

**Fig. 7 | The dipolar Josephson effect of planons. a** A proposed experimental scheme. In step 1, the system is initialized at the ground state of BHM in each chain. Different phases $\theta_1 \neq \theta_2$ are imprinted to these two chains. In step 2, the ring-exchange interaction is turned on at $\tau = 0$ and allows planons to tunnel between these two chains. In step 3, a supercurrent of dipoles between two chains is measured by tracing the dipole moment of each chain as time goes by. **b** Numerical results of the dipole moments of both chains $P_1$ and $P_2$ and the total dipole moment

$P$ as functions of the time. $t_1/U = 1$, $n_0 = 1$, $L = 3$ and $\Delta\theta \equiv \theta_1 - \theta_2 = \pi/2$. **c** Supercurrents of planons $J_D$ versus time for various $\Delta\theta$. **d** The dependence of $J_D(\tau = 0)$ on $\Delta\theta$. $t_1/U = 0.001$ and $L = 3$. Numerical simulations are shown by dots whereas curves are analytical results from Eq. (41) and Eq. (43). **e** The dependence of $J_D(\tau = 0)$ on $L$. $\Delta\theta$ has been chosen as $\pi/2$. When $t_1/U = \infty(0.1)$, numerical results agrees with Eq. (43) shown by red(orange) lines. The other two lines are linear fits of numerical results.

arbitrary plaquette between the two chains, accounting for the $L$-dependence of $J_D$ in Fig. 7e.

Whereas the above discussions apply to planons, Josephson effects also exist for lineons. Figure 8a shows a simple example of a three-site problem. Initially, the Hamiltonian is written as

$$H_i = -t_1\left(e^{i\theta_1} b_2^\dagger b_1 + e^{i\theta_2} b_3^\dagger b_2 + \text{h.c.}\right) + \frac{U}{2}\sum_{i=1}^{3} n_i(n_i - 1). \quad (44)$$

Although this is a small system with only 3 sites, we could still discuss macroscopic quantum phenomena and dipole condensates in the large $N$ limit, similar to the conventional double-well condensates[64,65]. Here, the second site divides the whole system into two parts, and a phase difference $\Delta\theta \equiv \theta_1 - \theta_2$ between the left half and the right half has been imprinted to the system. Since such a finite $\Delta\theta$ exists at the ground state of $H_i$, a supercurrent is absent initially. Now at time $\tau = 0$, we change the Hamiltonian to a three-site version of the DBHM

$$H_f = -t_2\left(b_2^{\dagger 2} b_3 b_1 + \text{h.c.}\right) + \frac{U}{2}\sum_{i=1}^{3} n_i(n_i - 1). \quad (45)$$

Without loss of generality, we define the dipole moment of each half of the system as

$$
\begin{aligned}
P_L(\tau) &= m n_1(\tau) + \frac{m+1}{2} n_2(\tau), \\
P_R(\tau) &= \frac{m+1}{2} n_2(\tau) + (m+2) n_3(\tau),
\end{aligned} \quad (46)
$$

where $m$ denotes the coordinate of the first site in an arbitrary reference frame. All following results are independent of $m$ and the choice of the reference frame. The total dipole moment is written as

$$P(\tau) = \sum_{i=1}^{3} (m + i - 1) n_i(\tau). \quad (47)$$

As shown in Fig. 8b, $P(\tau)$ is time-independent as $H_f$ conserves the total dipole moment. From $J_D = \partial P_L/\partial\tau$, we obtain the time-dependent supercurrent of dipoles. Again, Eq. (41) applies When $\tau \to 0$, and the constant $A$ can be derived in certain extreme cases. We have found that

$$A = \begin{cases} \frac{4}{3} t_2 \left(\frac{t_1}{U}\right)^2 n_0(n_0 + 1)(5n_0^2 + 5n_0 - 1), & \frac{t_1}{U} \ll 1, \\ \frac{3}{4} t_2 (3n_0^2 - n_0), & \frac{t_1}{U} \gg 1. \end{cases} \quad (48)$$

As shown in Fig. 8d, e, the analytical results shown by curves agree well with numerical simulations.

## Multipolar condensates

The above discussions about dipole condensates can be straightforwardly generalized to an arbitrary multipolar condensate. For instance, a quadrupole condensate is readily accessible once the DBHM in Eq. (1) or Eq. (2) is realized. In the limit where $t_2/U = 0$, the ground state of a DBHM is $|MI\rangle$. It has been shown there exists a critical value $(t_2/U)_c$ across which the ground state becomes a dipole condensate[21,22].

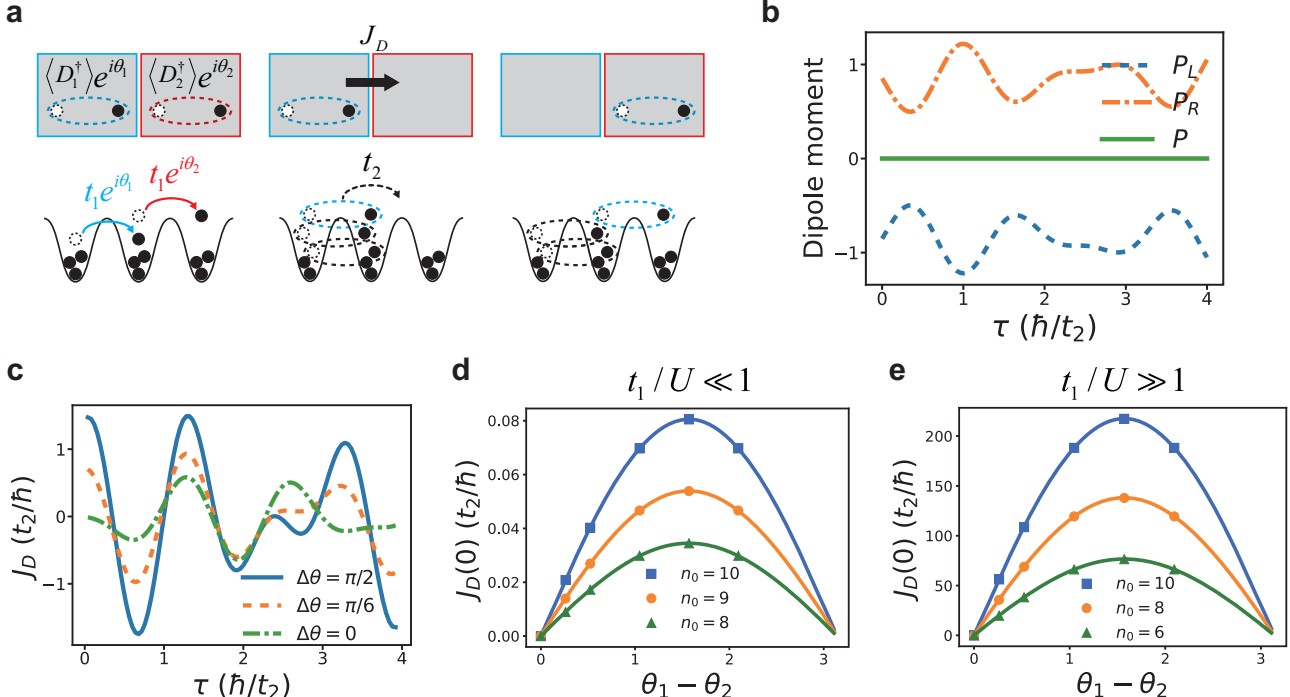

**Fig. 8 | The dipolar Josephson effect of lineons. a** An experimental proposal. In step 1, the system is initialized at the ground state of $H_i$ as shown in Eq. (44). A phase difference $\theta_1 - \theta_2$ is imprinted between the dipoles on the left half and right half. In step 2, the Hamiltonian is changed to $H_f$ as shown in Eq. (45). This allows lineons to tunnel but forbids single-particle tunnelings. In step 3, the supercurrent of dipoles is measured by tracing the dipole moments of the left and the right half of the system. **b** Results of the dipole moment on the left half ($P_L$) and right half ($P_R$) of the system, and the total dipole moment $P$ as functions of the time. $t_1/U = 1$, $n_0 = 1$, $L = 3$, and $\Delta\theta = \pi/2$. **c** Supercurrents $J_D$ versus time for various $\Delta\theta$. **d, e** $J_D(\tau = 0)$ versus $\Delta\theta$ at $t_1/U = 0.001$ (**d**) and $t_1/U = \infty$ (**e**). Numerical results are shown by dots whereas curves are obtained analytically from Eq. (41) and Eq. (48).

Though the dipole condensate vanishes when $t_2/U < (t_2/U)_c$, we need to emphasize that the ground state is not featureless. Similar to previous discussions about dipole condensates in BHM, here, a quadruple condensate exists when $t_2/U < (t_2/U)_c$. In the limit $t_2/U \to 0$, as analogous to Eq. (6), the ground state of the Hamiltonian in Eq. (1) is written as

$$|\Psi_2\rangle \approx |\text{MI}\rangle + \frac{t_2}{3U} \sum_i \left( b_i^{\dagger 2} b_{i-1} b_{i+1} + \text{h.c.} \right) |\text{MI}\rangle. \quad (49)$$

As a result of the self-proximity effect, the kinetic energy of the lineon automatically induces a quadrupole condensate on a 1D lattice.

In 2D, a quadrupole condensate also arises as the ground state of the Hamiltonian in Eq. (2) when $t'_2/U \to 0$,

$$|\Psi'_2\rangle \approx |\text{MI}\rangle + \frac{t'_2}{2U} \sum_{\mathbf{i}} \left( b_{\mathbf{i}+\hat{\mathbf{x}}}^{\dagger} b_{\mathbf{i}+\hat{\mathbf{y}}}^{\dagger} b_{\mathbf{i}} b_{\mathbf{i}+\hat{\mathbf{x}}+\hat{\mathbf{y}}} + \text{h.c.} \right) |\text{MI}\rangle. \quad (50)$$

Here the quadrupole condensate is induced by the kinetic energy of the planon, i.e., the ring-exchange interaction. Further controlling the phase of $t'_2$ will allow experimentalists to manipulate the phase of quadrupole condensates and thus deliver the quadrupole Josephson effect. Whereas the movement of a quadrupole could be more complex than a dipole, the generic principle is the same. After producing position-dependent phases, turning on the kinetic energy of the quadrupole leads to a supercurrent of quadrupoles.

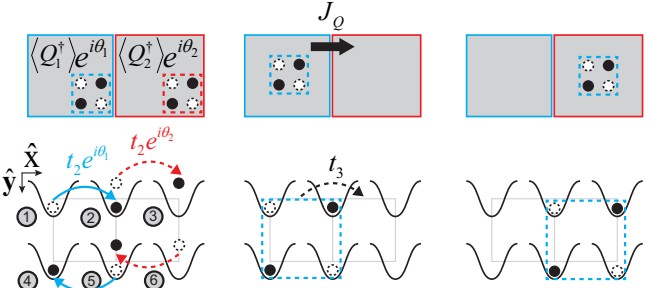

**Fig. 9 | The quadrupolar Josephson effect.** In step 1, the system is initialized at the ground state of the Hamiltonian in Eq. (51). A phase difference $\theta_1 - \theta_2$ is imprinted. In step 2, the Hamiltonian changes to Eq. (52) at $\tau = 0$ to allow quadrupoles to tunnel. In step 3, the supercurrent of quadrupoles is measured by tracing the quadrupole moment of each half of the system as time goes by.

Figure 9 shows an example of the quadrupole Josephson effect in a 6-site system. The initial Hamiltonian is written as

$$H_i = -\left( t_2 e^{i\theta_1} b_2^{\dagger} b_4^{\dagger} b_1 b_5 + t_2 e^{i\theta_2} b_3^{\dagger} b_5^{\dagger} b_2 b_6 + \text{h.c.} \right) + \frac{U}{2} \sum_{\mathbf{i}} n_{\mathbf{i}}(n_{\mathbf{i}} - 1). \quad (51)$$

When $t_2/U < (t_2/U)_c$, the ground state of $H_i$ is a quadrupole condensate, and $\theta_1(\theta_2)$ is the phase of quadrupoles on the left(right) plaquette. At time $\tau = 0$, the Hamiltonian is switched to

$$H_f = -\left( t_3 b_2^{\dagger 2} b_4^{\dagger} b_6^{\dagger} b_5^2 b_1 b_3 + \text{h.c.} \right) + \frac{U}{2} \sum_{\mathbf{i}} n_{\mathbf{i}}(n_{\mathbf{i}} - 1) \quad (52)$$

that turns on the quadrupole kinetics. As analogous to Eq. (41), the supercurrent of quadrupoles when $\tau \to 0$ follows the Josephson relation,

$$J_Q = A \sin(\theta_1 - \theta_2). \tag{53}$$

The above discussions apply to any multipolar condensates of arbitrary order. For instance, we consider a quadrupole Bose-Hubbard model,

$$H_3 = -\sum_{\mathbf{i}} \left( t_3 b^\dagger_{\mathbf{i+x}} b^\dagger_{\mathbf{i+y}} b^\dagger_{\mathbf{i+z}} b^\dagger_{\mathbf{i+x+y+z}} b_{\mathbf{i}} b_{\mathbf{i+x+y}} b_{\mathbf{i+x+z}} b_{\mathbf{i+y+z}} \right.$$
$$\left. + \text{h.c.} \right) + \frac{U}{2} \sum_{\mathbf{i}} n_{\mathbf{i}}(n_{\mathbf{i}} - 1). \tag{54}$$

When $t_3/U$ is larger than a critical value, a quadrupole condensate is expected as the ground state. For a generic value of $t_3/U$ less than this critical value, the quadrupole condensate is replaced by octapole condensate. More generically, a multipolar Bose-Hubbard model of the $n$-th order can be written as

$$H_n = -\sum_{\langle \mathbf{i}, \mathbf{j} \rangle} t_n M^\dagger_{n\mathbf{i}} M_{n\mathbf{j}} + \text{h.c.}, \tag{55}$$

where $M^\dagger_{n\mathbf{i}} = \prod_{j=1}^{2^{n-1}} b^\dagger_{\mathbf{i+k}_j} \prod_{j=1}^{2^{n-1}} b_{\mathbf{i+k}'_j}$ is a creation operator of a multipole of the $n$-th order, with a vanishing $m$-th order ($m < n$) multipole moment. When $t_n/U$ is smaller than a critical value, a multipolar condensate of $(n+1)$-th order arises as the ground state. Such multipolar Bose-Hubbard models thus provide us with a hierarchy of multipolar condensates.

## Experimental realizations

Many of our theoretical results are readily accessible using currently available experimental techniques. For instance, the BHM has been widely explored using ultracold atoms in optical lattices[45,47,66]. A Mott insulator with short-range correlations has also been observed in experiments[48]. Experimentalists could also increase the temperature to suppress the condensate and achieve a normal phase of bosons at finite temperatures in optical lattices. Therefore, a dipole condensate already exists in laboratories.

To access the dipolar Josephson effect, experimentalists need to perform the following few steps of experiments.

1. Manipulating the phase of a dipole condensate. In the first step, a position-dependent phase needs to be added to the tunneling $t_1$. By tilting an optical lattice, the bare tunneling is suppressed. Applying external lasers to overcome the energy mismatch between the nearest neighbor sites, the laser-induced tunneling naturally inherits the phase of the laser. In particular, following the protocol of two pioneering experiments[67,68], this phase can be made position-dependent such that the phase of the dipole condensate is twisted in real space. An extreme case is that the left half and the right half of the system have different phases. More generally, the phase of the dipole condensate may change smoothly, for instance, as a linear function of the lattice site.

2. Turning on correlated pair tunnelings. After imprinting a phase twist to the dipole condensate, the next step is to suppress the single-particle tunneling and introduce correlated pair hopping at the same time. This can be done by simply turning off the lasers that produce the laser-induced tunneling in the first step and meanwhile, keeping the tiled potential. As such, single-particle tunnelings are quenched and correlated pair tunnelings become crucial. The kinetic energy in Eq. (1) can be realized using similar schemes as in ref. 40,41. In addition, the ring-exchange interaction in Eq. (2) has been realized for fermions[39] and this scheme can be generalized to bosons. Applying linear field gradients along both directions of a square lattice, single-particle tunnelings are suppressed along both the $x$ and the $y$ directions. However, the

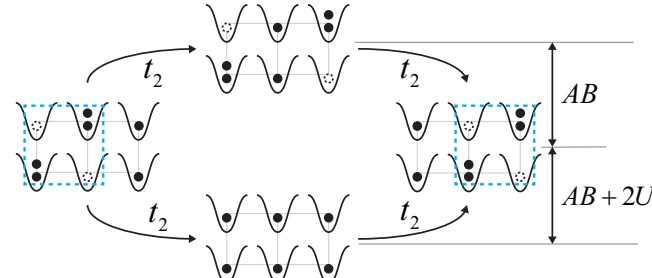

**Fig. 10 | The tunneling of a quadrupole.** With an additional quadratic potential $(Ax + By)^2/2$, a quadrupole could tunnel through second-order processes of the ring-exchange interaction.

ring-exchange interaction satisfies the resonant condition and a pair of bosons can then tunnel simultaneously, producing the ring-exchange interaction as shown in Fig. 1b. An alternative scheme is to implement long-range interactions. As discussed in a recent work[69], the long-range interaction between two parallel layers forces the particle in one layer and the hole in the other layer to tunnel in the same direction of each layer, resulting in the ring-exchange interaction. Such correlated pair tunnelings then produce currents of dipoles in the systems.

3. Measuring the time-dependent dipole moment of a subsystem. Following step 2, experimentalists could measure how the dipole moment of a chosen subsystem changes as a function of time. For instance, as shown in the previously discussed examples, the dipole moment of the left half (or the right half) of the system, $P_L(\tau)$ (or $P_R(\tau)$). From $\frac{dP_L(\tau)}{d\tau}$ (or $\frac{dP_R(\tau)}{d\tau}$), the current of dipole $J_D(\tau)$ could be obtained. Then $J_D(\tau)$ in the limit of $\tau \to 0$ could be compared with the phase twist of the dipole condensate, such as $\theta_1 - \theta_2$ in the previously discussed examples. Experimentalists then could measure the dipolar Josephson effect and test the predicted Josephson relation for dipole condensates.

To design the kinetics of quadruples and higher-order multipoles, extra work is required. For instance, we consider a 2D system in Fig. 9 and a Hamiltonian in Eq. (52) with an additional quadratic potential, $(Ax + By)^2/2$. The quadrupole conservation generated by a quadratic trap has been discussed in ref. 70. Here, this quadratic potential resembles a quadratic and a linear potential for single particles and dipoles, respectively, and can suppress their tunnelings. A quadruple, however, feels a constant potential and could tunnel without an energy penalty. As shown in Fig. 10, the initial state has a quadruple in the left half of the system and has an energy of $AB + 2U$ with respect to a Mott insulator with a unit filling. After turning on the ring-exchange interaction $t_2$, two pathways of second-order processes allow the quadrupole to move to the right half of the system. The final state has the same energy as the initial one. The tunneling amplitude of the quadrupole, $t_3$ in Eq. (52), is written as

$$t_3 = \frac{2Ut_2^2}{AB(AB + 2U)}. \tag{56}$$

Whereas the above process represents the nearest neighbor tunneling of the quadrupole, it is straightforward to verify that any long-range tunneling of the quadrupole is suppressed due to destructive interferences between different pathways. Such a scheme could also be generalized to higher dimensions to allow the quadrupole to move in more directions.

## Discussion

In this paper, we have shown that dipole condensates are as prevalent as conventional single-particle condensates. An ordinary normal phase of bosons, which is defined by a vanishing single-particle order

parameter, could readily be a dipole condensate. The single-particle kinetic energy automatically induces dipole condensation. Similarly, the kinetic energy of the $n$-th order multipoles produces a condensation of the $(n+1)$-th order multipoles. This can be understood as the self-proximity effect, resembling the conventional proximity effect, though no extra external condensates are required. Manipulating the phase of multipolar condensates will allow experimentalists to access a new type of Josephson effect, where the supercurrent of the $n$-th order multipoles arises in the absence of supercurrents of any $m$-th ($m < n$) order multipoles.

Whereas the dipole condensates and the dipolar Josephson effect are readily accessible in current experiments, it will be useful to realize even higher-order multipolar condensates and higher-order Josephson effects. Though the tunnelings of higher-order multipoles are, in principle, accessible via higher-order single-particle processes, this requires experimental advancement to realize such tunnelings in laboratories. In addition, it will be interesting to introduce higher rank tensor gauge fields and study their interactions with multipolar condensates. We hope that our work will stimulate more interest to study multipolar condensates in broader systems including but not limited to fermionic systems and systems with internal degrees of freedom.

### Reporting summary

Further information on research design is available in the Nature Portfolio Reporting Summary linked to this article.

## Data availability

Numerical data for the presented plots are available from the authors upon request.

## Code availability

Computer codes for generating the figures presented are available from the authors upon request.

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

## Acknowledgements

We thank helpful discussions with Senthil Todadri and Ethan Lake. This work is supported by The U.S. Department of Energy, Office of Science through the Quantum Science Center (QSC), a National Quantum Information Science Research Center.

## Author contributions

Q.Z. conceived the idea. W.X., C.L., and Q.Z. contributed to all aspects of this work. Q.Z. supervised the project.

## Competing interests

The authors declare no competing interests.
