## [Peer Review File · Nature Communications]

Multipolar condensates and multipolar Josephson effectsREVIEWER COMMENTS

Reviewer #1 (Remarks to the Author):

In this manuscript the authors discuss dipolar (and multipolar condensates) in tilted Bose-Hubbard (and related) models, paying special attention to their measurable properties (in particular Josephson effects), and also to routes to experimental realization. This strikes me as a timely and significant contribution to an exciting and rapidly developing field, likely to stimulate new experiments, and I am generally in favor of its publication.

One comment that I have regards referencing. The idea of generating dipole moment conservation from tilted potentials was first advanced, to the best of my knowledge, in Phys. Rev. B 101, 174204 (2020) [Section VI]. Meanwhile, dipole Fermi surfaces were discussed in Phys. Rev. B 97, 085116 (2018)[Section IV], whereas dipolar condensates were discussed in Phys. Rev. B 105, 155107 (2022). All of these papers precede the authors Refs 20,21, and I would encourage the authors to read them (and also to cite them as part of the background literature). Some of the ideas in the present manuscript have in fact already appeared in these early papers [e.g. the idea for generating quadrupole conservation using a quadratic trap] and it would be appropriate to acknowledge this. Nevertheless there is still plenty in the current manuscript that is new and significant, so this is not a complaint about novelty, merely a request that the authors clearly delineate the results that are novel from those that have already appeared in the literature.

My main other suggestion is with regards to the section on experimental realizations (Sec.VI). I think this is one of the most valuable parts of the paper - much of the physics under discussion is already accessible in cold atom experiments, and my hope is that this manuscript will stimulate new experiments exploring the effects that the authors identify. However, to maximize the chances of that good outcome, I think the authors should really expand the section on experimental realizations, clarifying what exactly are the experiments that should be done to see e.g. the multipolar Josephson effects, and what the key signatures would be. I don't expect specific laser frequencies or arrangements, but I would like to see a clearly laid out proposal for what experiment should be done and what is expected to be seen - the current discussion is a little too cryptic.

Once the above changes are made I would be happy to recommend this for publication.

Reviewer #2 (Remarks to the Author):

This manuscript proposes a novel concept of multipolar condensate, which is argued to generically arise in various types of bosonic systems. Authors find that dipole condensates prevail in conventional normal phases of bosons including finite-tunneling Mott insulators and finite-temperature Bose liquids. As one macroscopic consequence of the dipole condensation, they study multipolar Josephson effects. A self-proximity effect is proposed to induce multi-polar condensate in the ground state. For experimental realization, they also develop a non-equilibrium approach for state preparation. One important contribution of this work to the literature is to show the dipole condensate is quite generic for bosonic systems, which implies wide opportunities to study such phases in cold atom experiments. Before I recommend the paper for publication on Nature Communications, I would like to suggest authors to clarify on the following issues.

1. It is not quite clear to me whether the dipole condensate is a well-defined phase of matter. Phases of matter can be classified according to symmetry or topology in condensed matter physics. If two states cannot be adiabatically connected by a local unitary transformation (respecting symmetry), these two states belong to different phases. The dipole condensate presented here can be adiabatically connected to Mott insulating phase or the normal Bose gas. There is no distinction in the aspect of spontaneous symmetry breaking either. Then I wonder in what sense the dipole condensate is a sharply defined phase.

2. I would suggest authors specify the symmetry of their considered Hamiltonian, especially the DBHM. It seems this Hamiltonian has higher symmetry than the standard Bose-Hubbard model. Such symmetries might be related to the dipole conservation law. Is the dipole condensate defined according to the spontaneous symmetry breaking of certain continuous symmetry of the DBHM? This should be clarified.

3. The dipolar Bose-Hubbard model seems quite related to the constrained Bose-Hubbard model or constrained spin chain [e.g., arXiv:2205.07901, arXiv: 2210.11072]. I would suggest authors elaborate about the connections or the distinction.

4. In creating Bose metal with non-equilibrium dynamics, authors considered discrete Fourier transform. But it appears to this transformation is highly non-local for a large system. Then how to implement the non-local transformation seems missing in this paper. Authors may consider the scheme proposed at PRB 106, 134313 (2022), where the Fourier transform is constructed by a sequence of local Hamiltonians.

5. This study is motivated in the context of fracton phases. But the relation between the dipole condensate and the fracton phase is quite illusive. I wonder if a Mott insulator of a simple Bose-Hubbard model with finite tunneling is by itself a fracton phase. It is quite unlikely, although the Mott state is a dipole condensate according to the authors' definition. This needs more clarification.

Reviewer #3 (Remarks to the Author):

The authors discuss the concept of dipole condensates in strongly correlated systems and their potential applications in studying fracton phases of matter. The authors show that dipole condensates are prevalent in bosonic systems due to a self-proximity effect, where single-particle kinetics induces a finite order parameter of dipoles. The dipole condensation can occur in conventional normal phases of bosons and the preparation of dipole condensates in bosonic metal and band insulator are introduced. Then, methods are proposed to manipulate the phase of a dipole condensate and observe dipolar Josephson effects. Finally, the possibility of creating a generic multipolar condensate and the hierarchy of multipolar condensates that can lead to new macroscopic quantum phenomena are discussed.

The paper is interesting and well-written. Bose-Hubbard model (BHM) has been widely studied, but very few reports of dipole condensates are discussed in such generic model. I think this work could open a new avenue to explore fracton phase of matter in generic bosonic systems, which may attract much attention, especially for experimentalists.

After reading this manuscript, I have some following concerns.

(1) It is claimed that the existence of a dipole condensate does not require the dipole conservation law in BHM. The authors should show in detail the difference between the dipolar condensates in BHM without dipole conservation and the ones in dipolar BHM with dipole conservation, including the properties and the experiment requirements for realization.

(2) Can you show some connections between the self-proximity effect and the properties of DBHM discussed in [PRB 106, 064511(2022), PRB 107, 195132 (2023)]?

(3) Is Fig. 2 (a) an exact phase diagram or just a schematic for general cases? How to get this picture?

(4) To realize the correlated tunneling in Hamiltonian (1), tilted optical lattice should be used? Can it be realized in other way?

(5) To realize the correlated tunneling in Hamiltonian (2), ring exchange interaction should be used. How to realize this in BHM?

(6) To observe the dipolar Josephson effect, one should change the Hamiltonian to a DBHM as Hamiltonian (1) or (2). This is very important. How to suddenly change from a BHM with modulated tunneling to a DBHM (both for planons and lieons)?

(7) The generalization from 1D to 2D is straightforward. Can the dipole condensates and the corresponding phenomena occur in a 3D lattice model?

Minor typos:

(a) Page 9: Again, Eq. (41) applies When ... (When -> when)

(b) Page 11: The kinetic energy in Eq. (1) can be been ... (been should be deleted)

Reviewer #1 (Remarks to the Author):

Reviewer's comment

In this manuscript the authors discuss dipolar (and multipolar condensates) in tilted Bose-Hubbard (and related) models, paying special attention to their measurable properties (in particular Josephson effects), and also to routes to experimental realization. This strikes me as a timely and significant contribution to an exciting and rapidly developing field, likely to stimulate new experiments, and I am generally in favor of its publication.

One comment that I have regards referencing. The idea of generating dipole moment conservation from tilted potentials was first advanced, to the best of my knowledge, in Phys. Rev. B 101, 174204 (2020) [Section VI]. Meanwhile, dipole Fermi surfaces were discussed in Phys. Rev. B 97, 085116 (2018)[Section IV], whereas dipolar condensates were discussed in Phys. Rev. B 105, 155107 (2022). All of these papers precede the authors Refs 20,21, and I would encourage the authors to read them (and also to cite them as part of the background literature). Some of the ideas in the present manuscript have in fact already appeared in these early papers [e.g. the idea for generating quadrupole conservation using a quadratic trap] and it would be appropriate to acknowledge this. Nevertheless there is still plenty in the current manuscript that is new and significant, so this is not a complaint about novelty, merely a request that the authors clearly delineate the results that are novel from those that have already appeared in the literature.

Our reply

We thank the reviewer for the support of our manuscript. We also appreciate that the reviewer pointed out a few previous works to us. We do agree that these papers precede Refs 20,21 of the original manuscript and relevant to our work. We have added them in the references of the updated version of our manuscript. They now become Refs. [10,25,68] of the updated version. We have also added one sentence in the paragraph below Fig 10 and explicitly pointed out that Phys. Rev. B 101, 174204 (2020) discussed how to generate quadrupole conservation using a quadratic trap. They read, "The quadrupole conservation generated by a quadratic trap has been discussed in ref. [68]."

Reviewer's comment

My main other suggestion is with regards to the section on experimental realizations (Sec.VI). I think this is one of the most valuable parts of the paper - much of the physics under discussion is already accessible in cold atom experiments, and my hope is that this manuscript will stimulate new experiments exploring the effects that the authors identify. However, to maximize the chances of that good outcome, I think the authors should really expand the section on experimental realizations, clarifying what exactly are the experiments that should be done to see e.g. the multipolar Josephson effects, and what the key signatures would be. I don't expect specific laser frequencies

or arrangements, but I would like to see a clearly laid out proposal for what experiment should be done and what is expected to be seen - the current discussion is a little too cryptic.

Our reply

We thank the reviewer's valuable suggestion of expanding Sec.VI. Whereas experimental realizations were partially covered in Sec.IV, we do agree with the reviewer that we should clarify what exactly experimentalists should do to maximize the chances of good outcome. In the updated version of the manuscript, we have followed the reviewer's suggestion and clarified each step of the proposed experiments together with the expected observation in Sec.VI.

Reviewer's comment

Once the above changes are made I would be happy to recommend this for publication.

Our reply

We have made changes based on the reviewers' comments and suggestions. We hope that the reviewer will be satisfied by the updated version of the manuscript and will kindly support publication of it in Nature Communications.

Reviewer #2 (Remarks to the Author):

Reviewer's comment

This manuscript proposes a novel concept of multipolar condensate, which is argued to generically arise in various types of bosonic systems. Authors find that dipole condensates prevail in conventional normal phases of bosons including finite-tunneling Mott insulators and finite-temperature Bose liquids. As one macroscopic consequence of the dipole condensation, they study multipolar Josephson effects. A self-proximity effect is proposed to induce multi-polar condensate in the ground state. For experimental realization, they also develop a non-equilibrium approach for state preparation. One important contribution of this work to the literature is to show the dipole condensate is quite generic for bosonic systems, which implies wide opportunities to study such phases in cold atom experiments. Before I recommend the paper for publication on Nature Communications, I would like to suggest authors to clarify on the following issues.

Our reply

We thank the reviewer for agreeing that the concept of multipolar condensates we proposed is novel and our work "*implies wide opportunities to study such phases in cold atom experiments*".

Reviewer's comment

1. *It is not quite clear to me whether the dipole condensate is a well-defined phase of matter. Phases of matter can be classified according to symmetry or topology in condensed matter physics. If two states cannot be adiabatically connected by a local unitary transformation (respecting symmetry), these two states belong to different phases. The dipole condensate presented here can be adiabatically connected to Mott insulating phase or the normal Bose gas. There is no distinction in the aspect of spontaneous symmetry breaking either. Then I wonder in what sense the dipole condensate is a sharply defined phase.*

Our reply

We thank the reviewer for the question that allows us to clarify an important issue. The reviewer is absolutely correct that phases of matter are classified by symmetry or topology. A fundamental example that is relevant here is the superfluid or condensate in the Bose Hubbard model, where the spontaneous U(1) symmetry breaking distinguishes it from the normal phase. As such, the off-diagonal long-range order (ODLRO) leads to a finite single-particle order parameter $\langle b_i \rangle \neq 0$ of the superfluid or condensate, in sharp contrast to the normal phase where $\langle b_i \rangle$ vanishes.

Here, the dipole condensate is a well-defined phase with a well-defined order parameter. In the Bose Hubbard model, the dipole condensate is precisely the normal phase (including the Mott insulating phase at zero temperature with a finite single-particle tunneling strength), not that adiabatically connecting to the normal phase. Such a dipole condensate is characterized by a vanishing single-particle order parameter $\langle b_i \rangle = 0$ and finite two-particle order parameter $\langle b_i^\dagger b_j \rangle \neq 0$. It is distinguished from the superfluid or condensate in the sense that the U(1) symmetry is not broken and the single-particle order parameter vanishes.

We need to clarify that we are not saying that we have found a new phase on the phase diagram of the Bose Hubbard model. Instead, we want to point out something important that was overlooked in the literature. A vanishing single-particle order parameter $\langle b_i \rangle = 0$ does not mean that the normal phase is featureless. In the Bose Hubbard model, a finite two-particle order parameter $\langle b_i^\dagger b_j \rangle \neq 0$ at any finite temperatures $T \neq \infty$ and finite tunneling $t_1 \neq 0$ means that the normal phase is readily a dipole condensate. Therefore, the dipole condensate is very generic in bosonic systems and there are wide opportunities to study dipole condensates in cold atom experiments, as the reviewer has recognized.

Reviewer's comment

2. *I would suggest authors specify the symmetry of their considered Hamiltonian, especially the DBHM. It seems this Hamiltonian has higher symmetry than the standard Bose-Hubbard model. Such symmetries might be related to the dipole conservation law. Is the dipole condensate defined according to the spontaneous*

symmetry breaking of certain continuous symmetry of the DBHM? This should be clarified.

Our reply

Yes, DBHM has higher symmetry than the standard Bose-Hubbard model. In addition to the global U(1) symmetry that leads to the charge conservation, DBHM has the U(1) dipole symmetry that gives rise to the dipole conservation. This can be quickly seen from that adding a linearly increasing phase to the operators in DBHM does not change the Hamiltonian. As such, the dipole condensate phase in certain regime on the phase diagram of DBHMs arises from the spontaneous breaking of the U(1) dipole symmetry.

Meanwhile, we need to clarify that dipole condensate in BHM arises because of a different reason, since BHM does not have the dipole U(1) symmetry. As the reviewer has recognized, dipole condensate in our work arises from the self-proximity effect. The single-particle kinetic energy inevitably induces a finite order parameter of the dipole condensate, as a counterpart of the ordinary proximity effect. We have clarified this point in the paragraph above section III in the original manuscript. Indeed, to produce a condensate or superfluid in experiments, one does not need to rely on the spontaneous symmetry breaking at the ground state. The ordinary proximity effect is a well celebrated example. Here, the self-proximity effect could provide experimentalists wide opportunities to access dipole condensates and more broadly, generic multipolar condensates, as the reviewer recognizes.

In response to this comment, we have added a reference, which discussed the dipole U(1) symmetry and dipole conservation, and also added discussions in the paragraph below Eq.(2) about the symmetry of DBHM, which read, “Unlike BHM that has a global U(1) symmetry, DBHM in Eq.(1) has an additional dipole U(1) symmetry, leading to the dipole conservation law [25]. This can be seen from the fact that H_2 in Eq.(1) remains unchanged after adding a phase to the bosonic operator, $b_j \rightarrow b_j e^{i\alpha_j}$, where $\alpha_j = j\varphi$ changes linearly as a function of the lattice index j and φ is a constant. Similarly, H_2' in Eq.(2) is invariant if $b_j \rightarrow b_j e^{i\alpha_j}$ and $\alpha_j = j_x\varphi_x + j_y\varphi_y$, where φ_x and φ_y are constant. The dipole U(1) symmetry of DBHM provides much richer physics than BHM. It has been shown that DBHMs host some intriguing phases as the ground states in different parameter regimes [21, 22]. The spontaneous breaking of the dipole U(1) phase leads to a new phase of dipole condensate.”

Reviewer's comment

3. The dipolar Bose-Hubbard model seems quite related to the constrained Bose-Hubbard model or constrained spin chain [e.g., arXiv:2205.07901, arXiv:

2210.11072]. I would suggest authors elaborate about the connections or the distinction.

Our reply

The dipole Bose-Hubbard model is exactly the constrained Bose-Hubbard model. arXiv: 2210.11072 discussed special properties of dipole Bose-Hubbard model (or the constrained Bose-Hubbard model) in one dimension such as fractonic Luttinger liquids as counterparts of ordinary Luttinger liquids in Bose-Hubbard model. arXiv:2205.07901 discussed constrained spin chain model which is related to hard-core fermions with the multipole conservation laws. Whereas this work is also very interesting but it is not directly related to multipolar condensates we studied. To show our respect, we have added both references at the end of the paragraph containing Eq. (1) in the updated version of our manuscript. The descriptions of these references in Sec.I read, “Eq.(1) was also referred as to the constraint Bose-Hubbard model that supports fractonic Luttinger liquids [23]. Constrained spin chains were also studied in the literature [24].”

Reviewer’s comment

4. *In creating Bose metal with non-equilibrium dynamics, authors considered discrete Fourier transform. But it appears to this transformation is highly non-local for a large system. Then how to implement the non-local transformation seems missing in this paper. Authors may consider the scheme proposed at PRB 106, 134313 (2022), where the Fourier transform is constructed by a sequence of local Hamiltonians.*

Our reply

Using standard optical lattices, the discrete Fourier transform may require highly non-local operations that may be very challenging in a large system. However, with the recent experiment developments in a variety of platforms, discrete Fourier transform can be directly implemented. We have cited Refs. 46-50 in the original manuscript (Ref. 52-56 in the updated version).

For instance, in superconducting circuits and ion traps, all-to-all interactions have been realized such that the realization of non-local couplings is no longer an issue. Interactions between remote atoms can also be tailored using cavities. A pair of optical tweezers can be moved close to each other for establishing quantum correlations between atoms and then be separated far away while retaining the quantum correlations.

We thank the reviewer for pointing out PRB 106, 134313 (2022) to us. We agree that it will be nice if a sequence of local Hamiltonians could produce Fourier transform. We have cited this paper at the end of the paragraph containing Eq.(32).

Reviewer’s comment

5. *This study is motivated in the context of fracton phases. But the relation between the dipole condensate and the fracton phase is quite illusive. I wonder if a Mott insulator of a simple Bose-Hubbard model with finite tunneling is by itself a fracton phase. It is quite unlikely, although the Mott state is a dipole condensate according to the authors' definition. This needs more clarification.*

Our reply

The Mott state with a finite single-particle tunneling t_1 in a simple BHM could be regarded as a fracton phase. First, it is a dipole condensate, as the reviewer recognizes. Second, despite that the dipole conservation law is absent in BHM, the single-particle tunneling can still be suppressed by quantum or thermal fluctuations.

This is the reason that $\langle b_i^\dagger b_j \rangle$ decays exponentially with increasing $|i-j|$ in the Mott insulating regime. Whilst one may argue that in DBHM, the single-particle tunneling is strictly prohibited, we need to keep in mind that, in realistic systems in laboratories, single-particle tunneling exists. Whereas certain experimental techniques such as a tilted lattice could be used to suppress the single-particle tunneling for realizing DBHM, this is only an approximation at low energies. If we consider higher order terms, residual single-particle tunnelings will arise. Though this will not affect any low-energy physics or long-range correlations, single-particle kinetics do survive at short distance. For instance, no matter how strong the tilting is, the amplitude of Bloch oscillation of single-particles is still finite. Based on this consideration, a Mott state in a simple BHM could be regarded as a fracton phase.

For sure, we agree with the reviewer that the relation between the dipole condensate and the fracton phase needs to be further clarified. We have added discussion at the end of the last paragraph in section 2A, which read, "It is also worth pointing out that, though the theoretical definition of a fraction phase of matter relies on the dipole conservation law, the realization of such a phase in practice, may involve microscopic physics that breaks the dipole conservation law. For instance, a titled lattice suppresses the single-particle tunneling such that the dominant pair tunneling creates the DBHM as an effective theory at low energies. Nevertheless, the single-particle tunneling still exists no matter how strong the tilting is, though it does not affect low-energy physics. In this sense, the dipole condensate is a fracton phase of matter, regardless of whether it arises from DBHM or BHM. As previously explained, despite the absence of the dipole conservation law in BHM, single-particle tunneling is suppressed at large distance (or low energies) in the normal phase."

We have clarified all questions from the reviewer and made changes accordingly. We hope that the reviewer will be satisfied by the new version of the manuscript. We will appreciate it if the reviewer may kindly support publication of it in Nature Communications.

Reviewer #3 (Remarks to the Author):

Reviewer's comment

The authors discuss the concept of dipole condensates in strongly correlated systems and their potential applications in studying fracton phases of matter. The authors show that dipole condensates are prevalent in bosonic systems due to a self-proximity effect, where single-particle kinetics induces a finite order parameter of dipoles. The dipole condensation can occur in conventional normal phases of bosons and the preparation of dipole condensates in bosonic metal and band insulator are introduced. Then, methods are proposed to manipulate the phase of a dipole condensate and observe dipolar Josephson effects. Finally, the possibility of creating a generic multipolar condensate and the hierarchy of multipolar condensates that can lead to new macroscopic quantum phenomena are discussed.

The paper is interesting and well-written. Bose-Hubbard model (BHM) has been widely studied, but very few reports of dipole condensates are discussed in such generic model. I think this work could open a new avenue to explore fracton phase of matter in generic bosonic systems, which may attract much attention, especially for experimentalists.

Our reply

We thank the reviewer for stating that our “*paper is interesting and well-written*” and that our “*work could open a new avenue to explore fracton phase of matter in generic bosonic systems, which may attract much attention*”.

Reviewer's comment

After reading this manuscript, I have some following concerns.

(1) It is claimed that the existence of a dipole condensate does not require the dipole conservation law in BHM. The authors should show in detail the difference between the dipolar condensates in BHM without dipole conservation and the ones in dipolar BHM with dipole conservation, including the properties and the experiment requirements for realization.

Our reply

We thank the reviewer for raising this important question, which is related to the 2nd comment of reviewer #2. The microscopic origin of the dipole condensate is different in dipolar BHM compared to that in BHM. In dipolar BHM, there is dipole conservation, which comes from dipole U(1) symmetry. As such, a dipole condensate arises from the spontaneous breaking of the dipole U(1) symmetry. In BHM, the dipole U(1) symmetry is absent, and only charge conservation is present not the dipole conservation. Dipole condensate therefore does not arise from spontaneous breaking of the dipole U(1) symmetry. Instead, dipole condensate is induced by the self-proximity effect.

Despite that the microscopic physics is different, the physical observables are the same. In both cases, the dipole condensate is characterized by a vanishing single-particle order parameter $\langle b_i \rangle = 0$ and finite two-particle order parameter $\langle b_i^\dagger b_j \rangle \neq 0$.

In the original version of the manuscript, we have explained the difference between the dipole condensate in dipolar BHM and that in BHM in the last paragraph of section 2C. Nevertheless, we agree with the reviewer that we should further clarify the difference between the dipolar condensates in BHM without dipole conservation and the ones in dipolar BHM with dipole conservation. We have expanded the last paragraph of section 2C in the updated version of the manuscript. It reads, “As previously explained, DBHM is characterized by the dipole U(1) symmetry and thus has the dipole conservation law. Spontaneous symmetry breaking of the dipole U(1) symmetry provides DBHM with a rich phase diagram including a dipole condensate as the ground state. In contrast, the dipole conservation law is absent in BHM. A dipole condensate arises from a completely different microscopic mechanism, the self-proximity effect. Nevertheless, the off-diagonal long-range orders of these two cases are the same, for instance, $\langle b_i \rangle = 0$, $\langle b_i^\dagger b_{i+x} \rangle \neq 0$, and $\langle b_i^\dagger b_{i+y} \rangle \neq 0$. An important difference between these two cases is that the phase of a dipole condensate in BHM is fixed by the single-particle tunneling, unlike the ground state of DBHM where the spontaneous symmetry breaking allows an arbitrary phase of the dipole condensate. This is similar to an ordinary proximity effect, where the phase of the induced condensate or superfluid is determined by external conditions, not by spontaneous symmetry breaking. Here, controlling the phase of the single-particle tunneling provides experimentalists with a unique means to tune and twist the phase of a dipole condensate. Turning on correlated hoppings then gives rise to a dipolar Josephson effect. This will be discussed in section IV.”

Reviewer’s comment

(2) Can you show some connections between the self-proximity effect and the properties of DBHM discussed in [PRB 106, 064511(2022), PRB 107, 195132 (2023)]?

Our reply

As we explained in the reply to comment (1), the dipole condensate phase in DBHMs is related to the spontaneous symmetry breaking while the dipole condensate phase in BHM arises from a different mechanism, the self-proximity effect. The similarity is that both cases have the same order parameters, for instance, $\langle b_i \rangle = 0$ and $\langle b_i^\dagger b_j \rangle \neq 0$. A key difference is that the phase of the dipole condensate in BHM is fixed by the single-particle tunneling, unlike the arbitrary phase of the dipole condensate in

DBHM arising from the spontaneous symmetry breaking of the dipole $U(1)$ symmetry. As explained in the reply to comment (1), we have expanded the last paragraph of section 2C to clarify the similarities and differences between these two cases.

Reviewer's comment

(3) Is Fig. 2 (a) an exact phase diagram or just a schematic for general cases? How to get this picture?

Our reply

Fig. 2 (a) is a schematic of the phase diagram of BHM, which has been studied extensively in the literature. To obtain the accurate phase boundary, advanced numerical techniques like quantum Monte Carlo are required, for instance, please see Nature Physics 4, 617–621 (2008) and Nature Physics 6, 998–1004 (2010). We have added these two papers in the references and added a short description of the phase diagram in the paragraph containing Eq.(3), which reads “Fig.2(a) shows a schematic of the phase diagram of BHM. The accurate phase boundary could be obtained from advanced numerics [42,43]”

Reviewer's comment

(4) To realize the correlated tunneling in Hamiltonian (1), tilted optical lattice should be used? Can it be realized in other way?

Our reply

Tilted optical lattice is the most straightforward and the most commonly considered method in the literature to realize the correlated tunneling. A linearly tilted potential suppresses the single particle tunneling while dipoles see a constant potential. We have added Ref. 22 that discussed the physical realization in detail. There should be, in principle, other methods to suppress the single-particle tunneling. For instance, any models that deliver flat bands could fulfill this purpose and correlated tunnelling could arise from interactions. However, in generic cases, multiple correlated tunnelings may arise, unlike the tilted optical lattices which provide a cleaner set up where only desired short-range correlated tunnelings exist. We thank the reviewer for raising this interesting question and will explore other ways to realize correlated tunnelings in the future.

Reviewer's comment

(5) To realize the correlated tunneling in Hamiltonian (2), ring exchange interaction should be used. How to realize this in BHM?

Our reply

This is related to comment (4), which addresses a generic question of how to engineer correlated tunnelings. In Nature Physics 13, 1195–1200 (2017), which is now Ref. 37

in the updated version of the manuscript, the ring exchange interaction has been realized for fermions. The idea is to use field gradient to suppress the single-particle tunneling. Either interactions or multiple steps of single-particle tunnelings as high order processes could produce the ring exchange interaction. Similar schemes can be used to create ring exchange interaction for bosons. The PI's group recently proposed an alternative scheme using long-range interactions. Please see arXiv:2306.15663 for details. We have added this reference in the updated version of our manuscript and have added a short discussion in Sec.VI to guide readers to these references. These discussions read "An alternative scheme is to implement long-range interactions. As discussed in a recent work [67], the long-range interaction between two parallel layers forces the particle in one layer and the hole in the other layer to tunnel in the same direction of each layer, resulting in the ring exchange interaction".

Reviewer's comment

(6) To observe the dipolar Josephson effect, one should change the Hamiltonian to a DBHM as Hamiltonian (1) or (2). This is very important. How to suddenly change from a BHM with modulated tunneling to a DBHM (both for planons and lieons)?

Our reply

A BHM with position-dependent phase in the single-particle tunneling could be created in a tilted lattice. While the tilting suppresses the bare tunneling, Raman lasers could deliver photon-assisted single-particle tunneling to overcome the tilting induced energy mismatch. Moreover, the lasers could imprint a position dependent phase to such photon-assisted single-particle tunneling. This is how the Hofstadter-Harper model was realized in experiments.

To suddenly change such a BHM to a DBHM, experimentalists just need to turn off the Raman lasers. Without laser-assisted tunneling, single particles are no longer able to overcome the energy mismatch between adjacent sites and single-particle tunneling is suppressed. Interaction induced correlated tunnelings become dominant. In 1D, one of a pair of bosons in the same lattice site could tunnel to the left and the other tunnel to the right, giving rise to the DBHM for a lineon. Similarly, in 2D, a ring exchange interaction delivers the DBHM for a planon.

In response to this comment, we have expanded the session about the experimental realization. We have clarified each step experimentalists need to do for observing the dipolar Josephson effect. Please see the paragraphs in red in section VI about experimental realizations.

Reviewer's comment

(7) The generalization from 1D to 2D is straightforward. Can the dipole condensates and the corresponding phenomena occur in a 3D lattice model?

Our reply

Both the lineon and the planon are sub-dimensional particles, i.e., the motion of a dipole is either parallel to the polarization axis (lineon) or perpendicular to the polarization axis (planon). We thus expect that adding an extra dimension will not lead to qualitative differences in dipole condensates in 3D.

We hope that we have answered all questions from the reviewer. We will appreciate it if the reviewer may kindly support publication of the updated version of our manuscript in Nature Communications.

REVIEWER COMMENTS

Reviewer #2 (Remarks to the Author):

Authors have addressed the concerns I had in the last review, to my satisfaction. I now recommend the paper for publication on *Nature Communications*.

Reviewer #3 (Remarks to the Author):

The authors have carefully revised the manuscript according to all Reviewers' Reports. Most of the revisions are satisfactory. However, there are still several points should be addressed.

(1) For a simple Bose-Hubbard model in the form of Eq. (5), or a generalized one in the form of Eq. (36), according the Gutzwiller's Ansatz, the ground state for $|t_1/U| \ll 1$ is exactly a Mott insulator. Why the authors give the ground state as Eq. (6), which is a Mott insulator with dipolar excitation? The authors should give more details about this.

(2) The authors introduce a dipole operator, $D_{\{i,s\}} = b_{\{i\}}^{\{+\}} b_{\{i+s\}}$, which is actually a composite operator of creation operator times annihilation operator, like a particle-hole pair introduced in [Phys. Rev. Lett. 93, 120406 (2004)]. Can the dipolar condensate be understood as the particle-hole condensate of bosons? In further, it would be better to derive an effective Hamiltonian for the particle-hole pair operators (i.e. the dipole operators) in a function of $D_{\{i,s\}} = b_{\{i\}}^{\{+\}} b_{\{i+s\}}$ and their h.c.

(3) The authors propose to realize their model by using a titled Bose-Hubbard model, in which the single-particle hopping is suppressed by the titling potential. However, even in such a titled Bose-Hubbard model, when the bias matches the interaction, significant single-particle resonant tunneling takes place. Will the resonant tunneling affect the dipolar condensate?

(4) For the model (1), its naturally supports dimer condensate of nonzero $\langle (b_i)^2 \rangle$. What's connection and difference between the phases of dipole condensate and dimer condensate?

Reviewer #2 (Remarks to the Author):

Reviewer's comment

Authors have addressed the concerns I had in the last review, to my satisfaction. I now recommend the paper for publication on Nature Communications.

Our reply

We thank the reviewer for supporting the publication of our manuscript.

Reviewer #3 (Remarks to the Author):

Reviewer's comment

The authors have carefully revised the manuscript according to all Reviewers' Reports. Most of the revisions are satisfactory.

Our reply

We thank the reviewer for agreeing that our manuscript has been carefully revised and most of the revision are satisfactory.

Reviewer's comment

However, there are still several points should be addressed. (1) For a simple Bose-Hubbard model in the form of Eq. (5), or a generalized one in the form of Eq. (36), according the Gutzwiller's Ansatz, the ground state for $|t_1/U| \ll 1$ is exactly a Mott insulator. Why the authors give the ground state as Eq. (6), which is a Mott insulator with dipolar excitation? The authors should give more details about this.

Our reply

We thank the reviewer for the questions that allow us to further clarify a few important points. We agree with the reviewer that the Gutzwiller's ansatz is often used to study Bose-Hubbard model. Whereas it faithfully produces the zero-temperature phase diagram that qualitatively agree with accurate numerical results, as a variational method, it misses some important features. For instance, as the reviewer has pointed out, when $|t_1/U|$ is smaller than the critical value, Gutzwiller's ansatz predicts that the ground state is "perfect" Mott insulator, where not only long-range correlations vanish but also short-range correlations are absent, i.e., $\langle b_i^\dagger b_j \rangle \sim \delta_{i,j}$. This is because Gutzwiller's ansatz treats the long-range and short-range correlations on equal foot such that short-range correlations also immediately vanish once the long-range correlation vanishes. However, more careful analyzes show that, as long as the tunneling strength t_1 is finite, short-range correlations could still exist despite that the long-range correlation has disappeared. In fact, it is known that the correlation length that characterizes how $\langle b_i^\dagger b_j \rangle$ decays only gradually decreases with

decreasing $|t_1/U|$, see for example, Phys. Rev. B 40, 546. This corresponds to the dipolar excitation on top of the “perfect” Mott insulator for any finite t_1/U . The best example is the limit where $|t_1/U| \ll 1$ such that the first order perturbation theory works, and the ground state wavefunction is written as Eq. 6 in our manuscript. Such short-range correlations in the Mott insulating regime have been observed in a well-celebrated experiment, PRL.95,050404 (2005) (Ref. 48 in the updated manuscript). The experimental data of the momentum distribution has clearly show that short-range correlations exist in the Mott insulating regime.

In response to this comment, we have added a discussion following Eq. (6), which reads “Eq. (6) is obtained by regarding the tunneling term $t_1 b_i^\dagger b_j + h.c.$ as the perturbation to the interaction term with only the first order in t_1/U kept. This approach faithfully captures short-range correlations in the small t_1/U limit, unlike the Gutzwiller’s ansatz that leads to the immediate disappearance of both long-range and short-range correlations once t_1/U is smaller than the critical value $(t_1/U)_c$ separating the superfluid phase and Mott insulator. In reality, only when $t_1/U = 0$, the ground state becomes a perfect Mott insulator with vanishing short-range correlation. For any finite t_1/U , it is known and has been observed in experiments that Mott insulators do exhibit short-range correlations, which are captured by Eq. (6) in the limit $t_1/U \rightarrow 0$ [48].”

Reviewer’s comment

(2) *The authors introduce a dipole operator, $D_{\{i,s\}} = b_{\{i\}}^\dagger b_{\{i+s\}}$, which is actually a composite operator of creation operator times annihilation operator, like a particle-hole pair introduced in [Phys. Rev. Lett. 93, 120406 (2004)]. Can the dipolar condensate be understood as the particle-hole condensate of bosons? In further, it would be better to derive an effective Hamiltonian for the particle-hole pair operators (i.e. the dipole operators) in a function of $D_{\{i,s\}} = b_{\{i\}}^\dagger b_{\{i+s\}}$ and their h.c.*

Our reply

We thank the reviewer for pointing out Phys. Rev. Lett. 93, 120406 (2004) that studies BEC of particle-hole pairs in spin 1/2 fermionic atoms.

Yes, the dipolar condensate studied in our paper can be understood as the particle-hole condensate of bosons. In the study of the DBHM in Eq. (1), the effective Hamiltonian has indeed been derived in a function of $D_{i,s}$ following standard perturbation theory, which rewrites the kinetic energy including four operators in terms of the dipolar order parameter multiplied by two operators. This has been discussed in the Appendix B of Phys. Rev. B 107,195132(2023) (Ref. 22 in the manuscript). As for the much simpler BHM, the kinetic energy includes only two operators, $D_{i,s=1}$ and its h.c., a simple perturbation method is readily sufficient to

obtain the ground state wavefunction and correlations and there is no need to reformulate more sophisticated effective theories.

To show our respect, we have cited Phys. Rev. Lett. 93, 120406 (2004) below Eq. (1), which reads “Similar composite operators of particle-hole pairs were considered in fermionic systems [25].”

Reviewer’s comment

(3) The authors propose to realize their model by using a titled Bose-Hubbard model, in which the single-particle hopping is suppressed by the titling potential. However, even in such a titled Bose-Hubbard model, when the bias matches the interaction, significant single-particle resonant tunneling takes place. Will the resonant tunneling affect the dipolar condensate?

Our reply

We agree that, when the bias well matches the interaction, interaction assisted tunneling will take place. This will lead to very rich phenomena. For instance, the dynamics of particle-hole excitations (dipoles) in a 2D resonantly tilted BHM model have been discussed in arXiv:2311.05695. They studied two kinds of dipole excitations on a certain initial state and their anisotropic dynamics. In our paper, we consider the parameter regime away from such resonances. Since the interaction assisted tunneling becomes only relevant near the resonance, away from the resonance, it is safe to ignore such a tunneling and the DBHM serves as the effective theory for a titled Bose-Hubbard model.

In response to this comment, we have added a few sentences above Eq. (2), which reads “It is worth pointing out that when the tilting potential between the nearest neighbor sites matches the interaction, interaction-assisted single particle tunneling will take place, which leads to very rich phenomena [26]. In our paper, we consider an off-resonant tilted lattice such that tunnelings of single-particles are not relevant and Eq. (1) serves as the effective Hamiltonian.”

Reviewer’s comment

(4) For the model (1), its naturally supports dimer condensate of nonzero $\langle (b_i)^2 \rangle$. What's connection and difference between the phases of dipole condensate and dimer condensate?

Our reply

It is unclear to us why model (1) naturally supports dimer condensate of nonzero $\langle b_i^2 \rangle$. If we define such an order parameter, the kinetic energy in DBHM in Eq.(1) is going to be written as $\langle b_i^2 \rangle b_{i+1}^\dagger b_{i-1}^\dagger + h. c.$. This corresponds to a next nearest

neighbor tunneling induced by a dimer condensate, and also indicates that a onsite dimer order parameter will induce a finite non-local correlations. Whereas this sounds interesting but it is unclear whether this may really occur in reality, as it is not

obvious that how a mean-field approach based on $\langle b_i^2 \rangle b_{i+1}^\dagger b_{i-1}^\dagger + h.c.$ may produce a finite dimer order parameter $\langle b_i^2 \rangle$ in a self-consistent means. In fact, Phys. Rev. B 107,195132(2023) that studied the phase diagram of the DBHM did not find a dimer condensate but a dipolar condensate. It seems more naturally to consider a dipolar order parameter $\langle b_i^\dagger b_{i+1} \rangle$ as Phys. Rev. B 107,195132(2023) did.

We also want to call the reviewer's attention that a main result of our work about DBHM in Eq.(1) is the small t_2/U limit. Though the ground state may be considered as a trivial Mott insulator with no correlations, we point out that short-range correlations still exist for any finite t_2/U , such that the ground state is not featureless but an interesting quadrupole condensate. A perturbation method immediately show that $\langle D_i^\dagger D_j \rangle \neq 0$ when $|i - j| \rightarrow \infty$, but the off-diagonal long-range order of dimers $\langle b_i^{\dagger 2} b_j^2 \rangle$ remain zero. This is similar to the small t_1/U limit of the BHM where the ground state is not featureless neither but a dipolar condensate.

We hope that we have answered all questions from the reviewer. We will appreciate it if the reviewer may kindly support publication of the updated version of our manuscript in Nature Communications.

REVIEWERS' COMMENTS

Reviewer #3 (Remarks to the Author):

The authors have revised the manuscript according to my report. I am happy to recommend it for publication.

Reviewer #3 (Remarks to the Author):

Reviewer's comment

The authors have revised the manuscript according to my report. I am happy to recommend it for publication.

Our reply

We thank the reviewer for supporting the publication of our manuscript.